# Integrating Bayesian Network Structure into Residual Flows and Variational Autoencoders

**Jacobie Mouton**                                                                 *jmout123@gmail.com*
*Computer Science Division*
*Stellenbosch University*
*South Africa*

**Steve Kroon**                                                                    *kroon@sun.ac.za*
*Computer Science Division*
*Stellenbosch University*
and
*National Institute for Theoretical and Computational Sciences*
*South Africa*

**Reviewed on OpenReview:** *https://openreview.net/forum?id=OsKXlWamTQ*

## Abstract

Deep generative models have become more popular in recent years due to their scalability and representation capacity. Unlike probabilistic graphical models, they typically do not incorporate specific domain knowledge. As such, this work explores incorporating *arbitrary* dependency structures, as specified by Bayesian networks, into variational autoencoders (VAEs). This is achieved by developing a new type of graphical normalizing flow, which extends residual flows by encoding conditional independence through masking of the flow's residual block weight matrices, and using these to extend both the prior and inference network of the VAE. We show that the proposed graphical VAE provides a more interpretable model that generalizes better in data-sparse settings, when practitioners know or can hypothesize about certain latent factors in their domain. Furthermore, we show that graphical residual flows provide not only density estimation and inference performance competitive with existing graphical flows, but also more stable and accurate inversion in practice as a byproduct of the flow's Lipschitz bounds.

## 1   Introduction

Normalizing flows (NFs) (Rezende & Mohamed, 2015; Tabak & Turner, 2013) have proven to be a useful tool in many machine learning problems. Typically parameterized by neural networks, NFs represent complex probability distributions as bijective transformations of a simple base distribution, while tracking the change in density through the multivariate change-of-variables formula. Variational autoencoders (VAEs) (Kingma & Welling, 2014; Rezende et al., 2014) also provide a powerful framework for constructing deep latent variable models, and unlike bijective flows, allow practitioners to specify the dimensionality of the latent space. By positing and fitting a generic model of the data-generating process, VAEs allow one to generate new samples and reason probabilistically about the data and its underlying representation. Despite the success of VAEs, they typically use overly simple latent distributions, e.g. fully-factorized Gaussian distributions for both the prior and variational posterior. Subsequent work has explored incorporating more complex latent variable distributions and have shown that this results in improved performance. For example, NFs can be included as part of the VAE's encoder network (Kingma et al., 2016), entangling the latent variables non-linearly to obtain a richer class of approximate posteriors. The prior distribution can also be made more complex, for example by stacking layers of latent variables to create a hierarchical structure (Sønderby et al., 2016). This increases the flexibility of the true posterior, leading to improved empirical results (Kingma et al., 2016).

In contrast to data-driven deep generative models such as NFs and VAEs, Bayesian networks (BNs) model distributions as a structured product of conditional distributions, allowing practitioners to specify both expert knowledge and quantitative information in a simple and interpretable manner. The aim of this work is to combine the strengths of both: the simplicity and interpretability of BNs, and the scalability and representation capacity of deep generative models. Traditional VAEs, as well as those with flow-enriched inference networks and/or stacked layers of latent variables, do not allow one to directly control the dependence structure encoded by the model. We therefore propose an approach to incorporate *rich* conditional distributions for *arbitrary* dependency structures into VAEs. We do this by extending both the prior and inference network with *graphical* NFs (Wehenkel & Louppe, 2021; Weilbach et al., 2020), which allow one to encode an arbitrary BN structure into these distributions through the NF architecture.

The flow making up the prior of such a structured VAE will be used in both transformation directions—in the normalizing direction for density estimation and in the inverse direction for sample generation. However, existing graphical flows do not emphasize providing stable and efficient inversion. While NFs are theoretically invertible, stable inversion is not always guaranteed *in practice* (Behrmann et al., 2021): if the Lipschitz constant of the inverse flow transformation is too large, numerical errors may be amplified. To address this, our work proposes a new graphical flow that encodes domain knowledge from a BN into a *residual flow* (Chen et al., 2019). Residual flows ensure theoretical invertibility by imposing a Lipschitz bound on the transformation. Graphical residual flows (GRFs) encode a predefined dependency structure by masking the residual blocks' weights, and obtain stable inversion in practice as a byproduct of the Lipschitz constraint.

The contribution of this work is therefore two-fold.[1] First, we propose graphical residual flows as a graphical NF that can stably and efficiently be inverted in practice. We compare the GRF to existing approaches on both density estimation and inference tasks and confirm that this method yields competitive performance. Our model exhibits accurate inversion that is also more time-efficient than alternative graphical flows with similar task performance. GRFs are therefore an attractive alternative to existing approaches when a flow is required to perform reliably in both directions. Second, we propose a structured VAE, termed the structured invertible residual network (SIReN) VAE, that employs GRFs to encode a predefined dependency structure over the latent and observed variables. We identify posterior collapse (Razavi et al., 2019)—where some latent dimensions become inactive and are effectively ignored by the model—as an issue with SIReN-VAE, as this phenomenon is influenced by the encoded structure. We consider various existing techniques to alleviate this phenomenon, and show that they lead to improved performance. Finally, we empirically show this model's potential for better generalization in data-sparse settings, as well as its ability to provide more interpretable latent spaces when practitioners know or can hypothesize about latent factors in their domain.

## 2 Background & Related Work

### 2.1 Bayesian Networks

Let $P$ be the joint distribution over variables $\mathcal{X} = \{X_1, \ldots, X_D\}$. We say that $X_i$ and $X_j$ are *conditionally independent* given $X_k$ in $P$, if $P(X_i, X_j | X_k) = P(X_i | X_k) P(X_j | X_k)$. Now, let $\mathcal{G}$ be a DAG with vertices corresponding to elements of $\mathcal{X}$, and let $\mathrm{Pa}_{X_i}^{\mathcal{G}}$ denote the parent vertices of $X_i$ in $\mathcal{G}$. $P$ factorizes according to $\mathcal{G}$ if $P$ can be expressed as the following product of *conditional distributions* (Koller & Friedman, 2009): $P(\mathcal{X}) = \prod_{X_i \in \mathcal{X}} P_i(X_i | \mathrm{Pa}_{X_i}^{\mathcal{G}})$. In this setting, the BN graph $\mathcal{G}$ provides a compact encoding of various conditional independence assumptions about $P(\mathcal{X})$, as indicated by the absence of directed edges between certain vertices. We aim to capture these conditional independencies in the generative models we construct.

### 2.2 Normalizing Flows

An NF (Rezende & Mohamed, 2015; Tabak & Turner, 2013) consists of a simple base distribution and a bijective transformation that provides a mapping between this base distribution and a more complex data distribution. The incurred change in density is tracked via the change-of-variables formula. For density

---

[1] An implementation of the GRF and SIReN-VAE, as well as our experimental code, can be found at `https://gitlab.com/pleased/grf-and-siren-vae`.

estimation, an NF $F_{\mathbf{x}\to\boldsymbol{\epsilon}}(\cdot)$, is used to transform a sample from the data distribution, $\mathbf{x} \sim p$, to a sample from the base distribution, $\boldsymbol{\epsilon} \sim p_0$ (Papamakarios et al., 2017). The sample's density is then given by

$$\log p(\mathbf{x}) = \log p_0(F_{\mathbf{x}\to\boldsymbol{\epsilon}}(\mathbf{x})) + \log |\det (J_{F_{\mathbf{x}\to\boldsymbol{\epsilon}}}(\mathbf{x}))| \quad , \tag{1}$$

where $J_F(a)$ denotes the Jacobian of the transformation $F$ at $a$. For amortized variational inference, where the aim is to infer the latent posterior $\mathbf{z} \sim q(\cdot|\mathbf{x})$ for a given observation $\mathbf{x}$, the flow $F_{\boldsymbol{\epsilon}\to\mathbf{z}}(\cdot)$ instead models the generative direction of the mapping: it transforms a sample from the base distribution to a sample from the latent posterior (Kingma et al., 2016). The density of the generated sample is given by

$$\log q(\mathbf{z}|\mathbf{x}) = \log p_0(\boldsymbol{\epsilon}) - \log |\det (J_{F_{\boldsymbol{\epsilon}\to\mathbf{z}}}(\boldsymbol{\epsilon}))| \quad . \tag{2}$$

Flows are typically trained to implement a specific transformation direction, depending on the task. The name *normalizing flow* is in reference to the procedure of letting a variable 'flow' through a series of transformations that 'normalizes' a complex data distribution into a simpler known base distribution. We refer to a flow that instead transforms (samples from) the base distribution to (samples from) the data distribution as a *generative flow*.[2] The need for accurate and efficient inversion arises when using either $F_{\mathbf{x}\to\boldsymbol{\epsilon}}$ to generate new $\mathbf{x}$ from $\boldsymbol{\epsilon} \sim p_0$, or when using $F_{\boldsymbol{\epsilon}\to\mathbf{z}}$ to compute the density for a $\mathbf{z}$ not generated by the flow. In principle, the trained flow can be inverted either analytically (if possible) or by using numerical methods. However, inversion may be slow or numerically unstable unless suitable modelling choices are made.

Recent approaches to constructing NFs can be divided into two main categories: finite and continuous (or infinitesimal) flows. Finite flows (Tabak & Turner, 2013) create a complex bijective mapping by composing simpler transformations—which we refer to as *flow steps*—while continuous flows (Chen et al., 2018) define the flow transformation implicitly in terms of an ordinary differential equation (ODE). One type of finite flow ensures tractable computation of the Jacobian determinant by enforcing an *autoregressive* dependency structure over the variables such that the Jacobian is triangular. Another type of finite flow, known as a residual flow, applies the update $\mathbf{x}^{(t+1)} = \mathbf{x}^{(t)} + g_t(\mathbf{x}^{(t)})$ at each flow step $t$ and ensures invertibility and tractability by applying suitable restrictions to $g_t$. Since one type of residual flow, known as a *contractive* residual flow, forms the basis of our proposed graphical residual flow, we discuss it in more detail below.

### 2.2.1 Contractive Residual Flows

Behrmann et al. (2019) show how to construct a finite NF by changing the normalization scheme of a traditional residual network's weights. Consider a residual network (He et al., 2016), $F(\mathbf{x}) = (f_T \circ \ldots \circ f_1)(\mathbf{x})$, composed of blocks $\mathbf{x}^{(t+1)} := f_t(\mathbf{x}^{(t)}) = \mathbf{x}^{(t)} + g_t(\mathbf{x}^{(t)})$, with $\mathbf{x}^{(0)} = \mathbf{x}$. $F$ is invertible if all of its component transformations $f_t$ are invertible, which holds if all $g_t$ are *contractive*, i.e. $\mathrm{Lip}(g_t) < 1$, where $\mathrm{Lip}(\cdot)$ denotes the Lipschitz constant for a transformation. Behrmann et al. (2019) ensure $\mathrm{Lip}(g_t) < 1$ by implementing $g_t$ as a composition of activations $h$ with $\mathrm{Lip}(h) \leq 1$ (e.g. LipSwish (Chen et al., 2019)), and affine layers with weight matrices $W_i$ satisfying $||W_i||_s < 1$. Here, $|| \cdot ||_s$ is the spectral norm, which can be computed with a power iteration (Gouk et al., 2021). The spectral norm of $W_i$ is constrained to $[0, c]$ by normalizing $W_i$ as

$$\widetilde{W_i} = \min(c, ||W_i||_s) \cdot \frac{W_i}{||W_i||_s} \quad . \tag{3}$$

A (non-trivial) upper bound on the Lipschitz constant of a contractive residual block $g_t$ also implies Lipschitz bounds for $f_t$ and $f_t^{-1}$: the Lipschitz constant of the forward mapping, $\mathrm{Lip}(f_t)$, is upper bounded by $1 + \mathrm{Lip}(g_t)$, while the Lipschitz constant of the inverse mapping, $\mathrm{Lip}(f_t^{-1})$, is upper bounded by $1/(1-\mathrm{Lip}(g_t))$. This bi-Lipschitzivity is an attractive property for stable and efficient inversion (Behrmann et al., 2021). To allow scaling to higher dimensions, Chen et al. (2019) use a "Russian roulette" unbiased and tractable estimate for the Jacobian determinant. A drawback is that it has unpredictable time and memory usage.

### 2.2.2 Normalizing Flows with Graphical Structures

Incorporating the dependency structure of a BN into NFs has begun to receive attention in the past two years. For finite flows, Wehenkel & Louppe (2021) consider only autoregressive flows where each step can be

---

[2]'Normalizing flow' is widely used for flows in both directions, but we make this distinction for clarity.

constructed from a *normalizer* and a *conditioner* function. A normalizer applies a bijective transformation to its input and is partially parameterized by the output of the conditioner, which controls the dependencies between the variables such that the Jacobian of the flow step is triangular. They consider two specific normalizer functions. The first is the *affine* transformation: $f(\mathbf{x}|\mathbf{m}, \mathbf{s}) = \exp(\mathbf{s}) \odot \mathbf{x} + \mathbf{m}$, where $\odot$ denotes element-wise multiplication. Here, $(\mathbf{m}, \mathbf{s})$ is the output of the (arbitrarily complex) conditioner. To encode the dependency structure, the conditioner is constrained such that $(m_i, s_i)$ is only a function of $x_i$'s parents in the corresponding BN, denoted by $\mathbf{x}_{Pa(i)}$, for $i = 1, \ldots, D$. Second they consider the monotonic normalizer (Wehenkel & Louppe, 2019): $f_i(x_i|\mathbf{c}_i) = \int_0^{x_i} g_i(t, \mathbf{c}_i)\, dt + \beta_i(\mathbf{c}_i)$, where $\mathbf{c}_i$ is the conditioner output, and $g_i$ and $\beta_i$ are two neural networks with a strictly positive and a real scalar output, respectively. Although one could implement $D$ separate conditioner functions that each take a different subset of variables, $\mathbf{x}_{Pa(i)}$, as input, Wehenkel & Louppe (2021) choose to use only a single neural network. To ensure that the correct independencies are still maintained, they perform $D$ passes through this neural network, masking out those inputs during forward pass $i$ that are not in $\mathbf{x}_{Pa(i)}$.

Weilbach et al. (2020) propose incorporating a graphical structure into continuous NFs. They consider a neural ODE system (Chen et al., 2018), $\frac{d}{dt}\mathbf{x}_t = f(\mathbf{x}_t, t)$, where the layers of the neural network $f$ take the form $h(\mathbf{x}, t) = \tanh\{W\mathbf{x} \odot \eta_1(t)\} + \mathbf{b} \odot \eta_2(t)$, where the $\eta_i$ are time-dependent linear gating functions. They incorporate a graphical dependency structure by applying a binary adjacency matrix as mask to the weight matrix $W$. This mask ensures that output $i$ is dependent on only the inputs corresponding to $x_i$ and its parents in the corresponding BN at each layer in the network. A drawback of this masking approach is that it restricts the width of each layer of the neural network to be the same as the dimension of $\mathbf{x}$.[3]

## 2.3 Variational Autoencoders

Let $p_\theta(\mathbf{x}, \mathbf{z})$ be a deep latent variable model over observed variables $\mathbf{x}$ and latent variables $\mathbf{z}$, parameterized by neural networks with parameters $\theta$. Optimizing $\theta$ using maximum (marginal) likelihood estimation is intractable in this setting because one typically cannot easily compute the evidence, $p_\theta(\mathbf{x})$. By employing amortized variational inference, one can however maximize the evidence lower bound (ELBO):

$$p_\theta(\mathbf{x}) \geq \mathcal{L}_{\theta,\phi}^{\text{ELBO}}(\mathbf{x}) = \mathbb{E}_{\mathbf{z} \sim q_\phi} \left[\log p_\theta(\mathbf{x}, \mathbf{z}) - \log q_\phi(\mathbf{z}|\mathbf{x})\right] , \tag{4}$$

where the inference network with variational parameters $\phi$ outputs an approximation $q_\phi(\mathbf{z}|\mathbf{x})$ to the true posterior $p(\mathbf{z}|\mathbf{x})$. The resulting model, that simultaneously optimizes $\theta$ and $\phi$, is known as a variational autoencoder (VAE). The vanilla approach to constructing a VAE assumes that all the latent variables are independent in the prior and conditionally independent given $\mathbf{x}$ in the approximate posterior, and also uses simple distributions for the prior, likelihood and posterior:

$$\boldsymbol{\mu}, \log \boldsymbol{\sigma} = \text{EncoderNeuralNet}(\mathbf{x}; \phi)$$
$$q_\phi(\mathbf{z}|\mathbf{x}) = \mathcal{N}(\boldsymbol{\mu}, \text{diag}(\boldsymbol{\sigma}^2))$$

$$p(\mathbf{z}) = \mathcal{N}(\mathbf{0}, I)$$
$$\boldsymbol{\mu}, \log \boldsymbol{\sigma} = \text{DecoderNeuralNet}(\mathbf{z}; \theta)$$
$$p_\theta(\mathbf{x}|\mathbf{z}) = \mathcal{N}(\boldsymbol{\mu}, \text{diag}(\boldsymbol{\sigma}^2))$$

where DecoderNeuralNet and EncoderNeuralNet are neural networks parameterized by $\theta$ and $\phi$, respectively.

## 3 Graphical Residual Flows

Assume a contractive residual flow $F = f_T \circ \ldots \circ f_1$, where each residual block $g_t$, $t = 1, \ldots, T$, is a fully-connected neural network with a single hidden layer[4] and activation function $h(\cdot)$, where $\text{Lip}(h) \leq 1$:

$$\mathbf{x}^{(t)} := f_t(\mathbf{x}^{(t-1)}) = \mathbf{x}^{(t-1)} + \widetilde{W}_2 \cdot h(\widetilde{W}_1 \cdot \mathbf{x}^{(t-1)} + b_1) + b_2 . \tag{5}$$

Here, $\widetilde{W}_i$ indicates a normalized weight matrix as in Equation (3), such that $\text{Lip}(g_t) < 1$. Similar to the work of Wehenkel & Louppe (2021) and Weilbach et al. (2020), we can encode the graphical structure of a BN, by ensuring that output $i$ of each $f_t$ is only a function of those inputs corresponding to $x_i$ and its

---

[3]Weilbach et al. (2020) therefore suggest incorporating additional auxiliary variables.
[4]The rest of this discussion is easily extended to residual blocks with more hidden layers.

parents in the BN graph. This can be achieved by suitably masking the weight matrices of each of the above residual blocks before applying spectral normalization. Given a BN graph, $\mathcal{G}$, over the components of $\mathbf{x} \in \mathbb{R}^D$, let $W_i' = W_i \odot M_i$, $i = 1, 2$, be the new masked weight matrices where the $M_i$ are binary masks ensuring that component $j$ of the residual block's output is only a function of the inputs corresponding to $\{x_j\} \cup \mathrm{Pa}_{x_j}^{\mathcal{G}}$. The update to $\mathbf{x}^{(t-1)}$ in block $f_t$ is then defined as follows:

$$\mathbf{x}^{(t)} := \mathbf{x}^{(t-1)} + \widetilde{W_2'} \cdot h(\widetilde{W_1'} \cdot \mathbf{x}^{(t-1)} + b_1) + b_2 \ . \tag{6}$$

The masks are constructed according to a new variant of MADE (Germain et al., 2015) which allows one to encode not only autoregressive, but arbitrary graphical structures. To construct these masks, we assign a specific subset of variables to each unit in the neural network: input units are assigned their corresponding unit sets $\{x_i\}$ and output units are assigned the sets $\{x_i\} \cup \mathrm{Pa}_{x_i}^{\mathcal{G}}$. Each hidden unit is randomly assigned one of the sets associated with the input and output units. To ensure each output unit has at least one valid path connecting it to the input, we ensure that each of the sets $\{x_i\}$ $i = 1, \ldots, D$, is assigned to at least one unit in each hidden layer. See Figure 1 for an illustrated example. A mask is then constructed by zeroing out a weight between two units in successive layers if the set assigned to the unit in the later layer is not a superset of the set assigned to the unit in the earlier layer Our proposed masking scheme overcomes the shortcomings of those used by Wehenkel & Louppe (2021) and Weilbach et al. (2020) in that it requires only a single pass through the network and allows arbitrary hidden layer widths. Since we

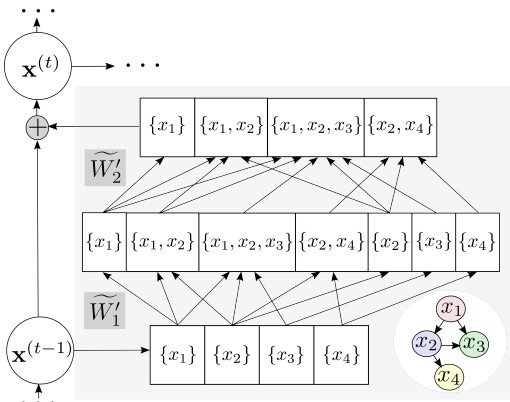

Figure 1: The update to $\mathbf{x}^{(t-1)}$ at flow step $t$ of a GRF. Edges removed by the masks are not shown. The remaining edges encode the structure of the given BN.

are enforcing the BN's DAG structure between the flow's variables, we are in effect encoding a 'sparse' autoregressive structure. If the variables were ordered according to their topological ordering in the BN, the update applied to each variable would only be conditioned on variables with a strictly lower index (though not necessarily on *all* variables with a lower index). Unlike standard residual flows, this construction results in a triangular Jacobian for which the determinant is easy to compute *exactly* as the product of the matrix's diagonal terms.

**Inversion** The inverse of this flow does not have an analytical solution (Behrmann et al., 2019). Instead, each block can be inverted numerically using a Newton-like fixed-point method (Song et al., 2019). To compute $\mathbf{x} = f_t^{-1}(\mathbf{y})$, the following update is applied until convergence:

$$\mathbf{x}^{(n+1)} = \mathbf{x}^{(n)} - \alpha \left( \mathrm{diag}(J_{f_t}(\mathbf{x}^{(n)})) \right)^{-1} \left[ f_t(\mathbf{x}^{(n)}) - \mathbf{y} \right] \ , \tag{7}$$

using the initialization $\mathbf{x}^{(0)} = \mathbf{y}$ and letting $0 < \alpha < 2$ (which ensures local convergence (Song et al., 2019)). We use this instead of the Banach fixed-point approach employed by Behrmann et al. (2019), since the convergence rate of the latter is dependent on the Lipschitz constant of the residual block and takes longer to converge for larger bounds. We empirically demonstrate that the convergence rate of (7) does not have this dependence (see Appendix B.3.1). It is thus preferable if we wish to use larger Lipschitz bounds for the residual blocks, which allows for more expressive flow steps. Although (7) only guarantees local and not global convergence like the Banach approach, we did not find this to be an issue with the datasets used.

**Induced Dependency Structure** For a single-step normalizing GRF, the conditional independencies of the distribution learned by the flow will correspond exactly to the conditional independencies specified by the encoded BN (Wehenkel & Louppe, 2021). When multiple flow steps are composed, additional dependencies are however induced—in a multi-step GRF each variable will ultimately receive information from all of its ancestors in the BN via the intermediate representations of its parents at previous steps, with the induced dependency structure corresponding to the BN's transitive closure. Because this 'information leakage' is

only through intermediate representations of each variable's parents, we nonetheless still expect GRFs to incorporate a strong enough inductive bias to encourage the variables to adhere to the desired dependency structure. See Appendix A.3 and B.3.3 for details and empirical results supporting this argument.

**Variational Inference**  In the case where latent variables $\mathbf{z}$ are present, one typically only has access to the forward BN that models the generating process for an observation $\mathbf{x}$. That is, the BN generally encodes the following generic factorization of the joint: $p(\mathbf{x}, \mathbf{z}) = p(\mathbf{x}|\mathbf{z})p(\mathbf{z})$. To perform inference, we first invert the BN structure using the faithful inversion algorithm of Webb et al. (2018) to obtain a BN structure encoding domain knowledge about the dependencies between the components of $\mathbf{z}$ once $\mathbf{x}$ is observed. This allows us to construct a GRF over the latent variables that is conditioned on the observation: each block is constructed as $\mathbf{z}^{(t)} := \mathbf{z}^{(t-1)} + \widetilde{W}_2' \cdot h(\widetilde{W}_1' \cdot (\mathbf{z}^{(t-1)} \oplus \mathbf{x}) + b_1) + b_2$, where $\oplus$ denotes concatenation. Such a GRF is typically applied in the *generative* direction to infer the latent representation of a given observation.

**LipMish Activation Function**  Since Equations (1) and (2) contain the derivatives of the residual block activation functions through the Jacobian term, the gradients used for training will depend on their second derivatives. It is thus desirable to use smooth non-monotonic functions that adhere to the Lipschitz bounds (such as LipSwish (Chen et al., 2019)), because unlike common monotonic activations, these typically do not have a vanishing second derivative in the region where the first derivative approaches 1. In our model, we use an activation we call "LipMish": $\mathrm{LipMish}(x) = (x/1.088) \cdot \tanh(\mathrm{softplus}(\mathrm{softplus}(\beta) \cdot x))$, which is a scaled version of the non-monotonic Mish function (Misra, 2020), ensuring that $\mathrm{Lip}(\mathrm{LipMish}) \le 1$ for all $\beta$, where $\beta$ allows for different degrees of curvature and is passed through $\mathrm{softplus}(\cdot)$ to ensure it is nonnegative.

## 4 VAEs with Structured Invertible Residual Networks

If we wish to construct a VAE, and have prior knowledge about the data-generating process, then it seems beneficial to incorporate this knowledge in the VAE. In this work, we assume access to a BN specifying the dependency structure over $D$ observed and $K$ latent variables. Our goal is to suitably incorporate this information into the VAE's encoder and decoder networks. Using $\theta$ for the decoder network's parameters, this means that its likelihood $p_\theta(\mathbf{x}|\mathbf{z})$ and prior $p_\theta(\mathbf{z})$ should ideally factorize according to the BN's conditional independencies. However, Webb et al. (2018) also showed the value of encoding the generative model's true inverted dependency structure as far as possible in the VAE's inference network. That is, the structure of the variational posterior should respect knowledge about $p(\mathbf{z}|\mathbf{x})$ which can be deduced from the factorization of $p(\mathbf{x}, \mathbf{z})$. Approximating the posterior $p(\mathbf{z}|\mathbf{x})$ in such a way requires inverting the BN (so that edges go from $\mathbf{x}$ to $\mathbf{z}$) while taking into account the independencies encoded by the model. As discussed in the context of inference with flows, we use Webb et al. (2018)'s proposed algorithm for obtaining such a *faithful* inverse.

We use GRFs to incorporate these desired structures into the generative and inference network of a VAE, yielding the structured invertible residual network (SIReN) VAE. For an observed sample $\mathbf{x}$, the encoder network (below left), with parameters $\phi$, is defined as a GRF conditioned on $\mathbf{x}$. The subscript $g$ denotes that this is a generative flow, and for all our investigations, we set $p_0$ to $\mathcal{N}(\mathbf{0}, I_K)$. For a sample $\mathbf{z}$ from the encoder network, the decoder (below right) uses a normalizing flow for the prior density, and a fully-factored Gaussian likelihood, with parameters output by a network denoted by DecoderNN:

$$\mathbf{z} = \mathrm{GRF}_g(\boldsymbol{\epsilon}; \mathbf{x}, \phi) \quad \text{where} \quad \boldsymbol{\epsilon} \sim p_0$$
$$\log q_\phi(\mathbf{z}|\mathbf{x}) = \log p_0(\boldsymbol{\epsilon}) - \log\left|\det(J_{\mathrm{GRF}_g}(\boldsymbol{\epsilon}))\right|$$

$$\log p_\theta(\mathbf{z}) = \log p_0(\mathrm{GRF}_n(\mathbf{z}; \theta)) + \log|\det(J_{\mathrm{GRF}_n}(\mathbf{z}))|$$
$$\boldsymbol{\mu}, \log\boldsymbol{\sigma} = \mathrm{DecoderNN}(\mathbf{z}; \theta)$$
$$p_\theta(\mathbf{x}|\mathbf{z}) = \mathcal{N}(\boldsymbol{\mu}, \mathrm{diag}(\boldsymbol{\sigma}^2))$$

The subscript $n$ above denotes a normalizing flow. To encode the conditional independencies between the latent and observed variables, DecoderNN is also masked according to the scheme discussed for GRFs. Note that $\mathrm{GRF}_n$ must be inverted to generate samples from this VAE, making sample generation slower than with regular VAEs. A key benefit of GRFs over other graphical flows, however, is that they guarantee stable inversion. The inversion time per flow step of a GRF is also relatively low compared to other graphical flows with similar modelling capability. This motivates our use of GRFs instead of other existing graphical flows.

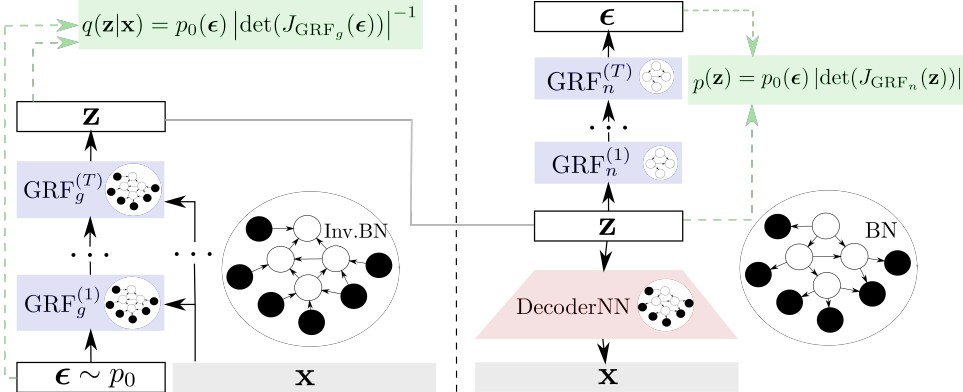

Figure 2: SIReN-VAE encodes the BN's graphical structure into the decoder (right) via masking of the normalizing GRF (GRF$_n$) and decoder neural network (DecoderNN) weights. The inference network (left) similarly encodes the inverted BN structure in its generating GRF (GRF$_g$).

## 5 Experiments

We evaluate the proposed GRF and SIReN-VAE models on a range of synthetic and real-world datasets that each have an associated true or hypothesized BN graph. The synthetic datasets are generated from fully specified BNs. All models were trained using the Adam optimizer with an initial learning rate of either 0.01 or 0.001 and a batch size of 100. The learning rate was decreased by a factor of 10 each time no improvement in the loss was observed for a set number of consecutive epochs, until a minimum learning rate of $10^{-6}$ was reached, at which point training was terminated. The initial learning rate and duration before learning rate reduction was chosen based on the lowest validation loss obtained over the grid $\{0.01, 0.001\} \times \{10, 20, 30\}$.

### 5.1 Graphical Residual Flows

We first compare GRF to two existing approaches—the graphical NF (GNF) of Wehenkel & Louppe (2021), for which we consider both affine and monotonic normalizers (denoted by GNF-A and GNF-M, respectively) and the structured conditional continuous NF (SCCNF) presented in Weilbach et al. (2020). Our experiments use three synthetic datasets: the Arithmetic Circuit dataset (Weilbach et al., 2020; Wehenkel & Louppe, 2021), an adaptation of the Tree dataset (Wehenkel & Louppe, 2021) as well as a linear Gaussian BN, EColi, adapted from the repository of Scutari (2022). We also consider two real-word datasets, namely Protein (Sachs et al., 2005) and MEHRA (Vitolo et al., 2018). Further information is given in Appendix B.1. To provide more informative comparisons between the flows, we train two models per task for each approach. The first is a smaller model with a maximum capacity of 5000 trainable parameters, denoted by a subscript S, e.g., GRF$_S$. We also train a larger model with a maximum capacity of 15000 parameters, denoted by the subscript L. For further information on the flow architectures, see Appendix B.2. Our experiments consider the relative performance of the flows with respect to tasks in both the generative and normalizing directions, as well as the efficiency and accuracy of flow inversion.

### 5.1.1 Density Estimation & Inference

Table 1 provides the negative log-likelihood (NLL) and the negative ELBO achieved by each model on the test set of the various datasets for the density estimation and variational inference tasks, respectively. For density estimation we assumed all variables were observed. Inference is only performed for the synthetic datasets, since we require access to the true model $p(\mathbf{x}, \mathbf{z})$ to compute the ELBO. We find that GRFs provide performance competitive with GNF-M and SCCNF, with the GRF models achieving the best NLL and ELBO on the majority of the datasets. GNF-A, with its reliance on simple affine transformations, is unable to match the performance of the other approaches for these primary modelling tasks.

Table 1: Density estimation and inference performance. NLL and $-$ELBO are the averages on the test set (lower is better) over five runs, with standard deviation given in the subscript. Bold indicates the best result in each group. The number of edges (E) and observed (D) and latent (K) variables in the datasets' associated BNs are also provided. Inference was not performed on the real-world datasets (where $K = 0$).

| | | | | | Density Estimation (NLL) | | Variational Inference ($-$ELBO) | |
|---|---|---|---|---|---|---|---|---|
| BN | D | K | E | Flow | Small Budget | Large Budget | Small Budget | Large Budget |
| Arithmetic Circuit | 2 | 6 | 8 | GNF-A | $1.26_{\pm 0.02}$ | $1.41_{\pm 0.16}$ | $4.90_{\pm 0.79}$ | $4.59_{\pm 0.28}$ |
| | | | | GNF-M | $1.19_{\pm 0.07}$ | $1.14_{\pm 0.04}$ | $\mathbf{3.96_{\pm 0.19}}$ | $3.92_{\pm 0.08}$ |
| | | | | SCCNF | $\mathbf{0.86_{\pm 0.01}}$ | $\mathbf{0.85_{\pm 0.00}}$ | $4.01_{\pm 0.07}$ | $3.97_{\pm 0.03}$ |
| | | | | GRF | $1.25_{\pm 0.01}$ | $1.11_{\pm 0.01}$ | $4.19_{\pm 0.19}$ | $\mathbf{3.71_{\pm 0.14}}$ |
| Tree | 1 | 6 | 8 | GNF-A | $9.32_{\pm 0.00}$ | $9.32_{\pm 0.00}$ | $2.36_{\pm 0.04}$ | $2.38_{\pm 0.05}$ |
| | | | | GNF-M | $8.65_{\pm 0.01}$ | $8.65_{\pm 0.01}$ | $\mathbf{1.72_{\pm 0.01}}$ | $\mathbf{1.70_{\pm 0.00}}$ |
| | | | | SCCNF | $\mathbf{8.59_{\pm 0.01}}$ | $\mathbf{8.59_{\pm 0.00}}$ | $1.78_{\pm 0.01}$ | $1.76_{\pm 0.01}$ |
| | | | | GRF | $8.64_{\pm 0.01}$ | $8.64_{\pm 0.00}$ | $1.74_{\pm 0.00}$ | $1.71_{\pm 0.00}$ |
| Protein | 11 | 0 | 20 | GNF-A | $6.93_{\pm 0.90}$ | $6.92_{\pm 0.57}$ | — | — |
| | | | | GNF-M | $-3.00_{\pm 0.77}$ | $-5.48_{\pm 0.23}$ | — | — |
| | | | | SCCNF | $-4.88_{\pm 0.21}$ | $-5.60_{\pm 0.05}$ | — | — |
| | | | | GRF | $\mathbf{-5.26_{\pm 0.01}}$ | $\mathbf{-6.11_{\pm 0.01}}$ | — | — |
| EColi | 29 | 15 | 59 | GNF-A | $40.11_{\pm 0.01}$ | $40.11_{\pm 0.00}$ | $34.98_{\pm 0.00}$ | $34.98_{\pm 0.00}$ |
| | | | | GNF-M | $40.13_{\pm 0.01}$ | $40.13_{\pm 0.00}$ | $34.99_{\pm 0.01}$ | $34.98_{\pm 0.01}$ |
| | | | | SCCNF | $40.12_{\pm 0.02}$ | $40.08_{\pm 0.01}$ | $35.24_{\pm 0.01}$ | $35.24_{\pm 0.01}$ |
| | | | | GRF | $\mathbf{40.06_{\pm 0.00}}$ | $\mathbf{40.06_{\pm 0.00}}$ | $\mathbf{34.96_{\pm 0.00}}$ | $\mathbf{34.96_{\pm 0.00}}$ |
| MEHRA | 10 | 0 | 10 | GNF-A | $12.90_{\pm 0.02}$ | $12.93_{\pm 0.03}$ | — | — |
| | | | | GNF-M | $11.74_{\pm 0.02}$ | $11.67_{\pm 0.02}$ | — | — |
| | | | | SCCNF | $11.80_{\pm 0.05}$ | $11.76_{\pm 0.02}$ | — | — |
| | | | | GRF | $\mathbf{11.66_{\pm 0.02}}$ | $\mathbf{11.61_{\pm 0.03}}$ | — | — |

We also compared the density estimation performance of GRF to the vanilla residual flow (RF) of Chen et al. (2019). The results are given in Table 2. The chosen RF architectures are similar to those of the GRFs they are being compared against: only the residual block hidden layer widths were reduced to ensure that the models comply with the respective size budgets. GRF performs the best on the majority of the datasets, specifically the synthetic datasets where the true dependency structure is known, but also on the real-world Protein dataset in the large model setting. We also found that GRFs typically required less time to perform density estimate calculations than RFs, which allowed for faster training. This is to be expected, given that the Jacobian determinant for the GRFs can be computed directly, whereas RFs must use the Russian roulette estimator. However, GRF is noticeably outperformed by RF on the real-world MEHRA dataset. Note that GRF has the best performance of all the graphical flows on this dataset, suggesting that its poorer performance may be a result of the BN not fully capturing the dependencies in the data, rather than a shortcoming of GRF. The clear superiority of RF over the graphical flows may cause one to question the veracity of the assumed BN structure. In such a case, the difference in results between vanilla RFs and GRFs can serve as a prompt to further investigate and refine the BN structure.

Table 2: NLL achieved by GRF compared to the vanilla residual flow (RF) of Chen et al. (2019).

| BN | Flow | Small Budget | Large Budget |
|---|---|---|---|
| Arithmetic Circuit | GRF | $\mathbf{1.25_{\pm 0.01}}$ | $\mathbf{1.11_{\pm 0.01}}$ |
| | RF | $1.27_{\pm 0.07}$ | $1.20_{\pm 0.03}$ |
| Tree | GRF | $\mathbf{8.64_{\pm 0.01}}$ | $8.64_{\pm 0.00}$ |
| | RF | $8.66_{\pm 0.02}$ | $\mathbf{8.62_{\pm 0.04}}$ |
| Protein | GRF | $-5.26_{\pm 0.01}$ | $\mathbf{-6.11_{\pm 0.01}}$ |
| | RF | $-5.78_{\pm 0.13}$ | $-5.84_{\pm 0.15}$ |
| EColi | GRF | $\mathbf{40.06_{\pm 0.00}}$ | $\mathbf{40.06_{\pm 0.00}}$ |
| | RF | $40.30_{\pm 0.08}$ | $40.42_{\pm 0.12}$ |
| MEHRA | GRF | $11.66_{\pm 0.02}$ | $11.61_{\pm 0.03}$ |
| | RF | $\mathbf{8.68_{\pm 0.06}}$ | $\mathbf{8.67_{\pm 0.08}}$ |

Table 3: Inversion performance on 100 test samples from the Tree dataset. Bold indicates the best results in each column. $N$ and $\alpha$ are not applicable for SCCNF. Ranges indicate different optimal settings for $N$ and $\alpha$ for different samples. Inversion time is measured for the smallest $N \leq 50$ that allowed the most samples in the batch to have a reconstruction error of less than $10^{-4}$, and is the time taken to invert the entire batch.

| | Small Budget | | | | Large Budget | | | |
|---|---|---|---|---|---|---|---|---|
| Flow | Converged within 50 steps | $N$ | $\alpha$ | Inversion time (ms) | Converged within 50 steps | $N$ | $\alpha$ | Inversion time (ms) |
| GNF-M | **100** | 4–9 | 0.9–1.0 | 53.67 | 98 | 4–47 | 0.4–1.0 | 488.51 |
| SCCNF | 98 | — | — | 140.51 | 94 | — | — | 392.97 |
| GRF | **100** | 3–5 | 1.0 | **49.73** | **100** | 4-6 | 0.9–1.0 | **122.27** |

### 5.1.2 Inversion

The main reason for introducing GRFs is their potential to provide more accurate inversion. Having established that they provide competitive key task performance, we now investigate the inversion accuracy and efficiency of GRFs compared to alternative graphical flows with similar task performance, namely GNF-M and SCCNF. GNF-M and GRF were inverted using the Newton-like inversion procedure in (7), while SCCNF was inverted by executing the integration in the opposite direction. For GNF-M and GRF, we considered values for the step-size $\alpha$ in the set $\{0.1 \times t \,|\, t = 1, ..., 19\}$, and the inversion process was deemed to have converged when a reconstruction error of less than $10^{-4}$ was achieved. To better illustrate potential inversion instability, we performed this inversion on a per-data-point basis for 100 test samples from the Tree dataset; the results for the other datasets are given in Appendix B.3.1. For each data point, we note the $\alpha$ that required the fewest iterations, $N$, for convergence. Table 3 summarizes these results, including the number of samples for which the desired reconstruction error was achieved for GNF-M and GRF with $N \leq 50$. We also include the inversion time for the entire batch, using the settings that allowed the most samples to achieve the desired reconstruction error.

One of the main paradigms for enforcing global inversion stability is using Lipschitz-constrained flow transformations (Behrmann et al., 2021). For GRFs, this stability is automatically achieved as a byproduct of the flow design, and we see that GRF shows excellent inversion accuracy. GNF-M, depending on the architecture and learned weights, has either potentially very large Lipschitz bounds, or no global bounds at all. This helps to explain its poorer inversion results. As illustrated in Figure 3, numerically inverting GNF-M can lead to very large reconstruction errors. While SCCNFs have global Lipschitz bounds, these are not controlled during training and numerical instability can occur. For further information see Appendix A.2.

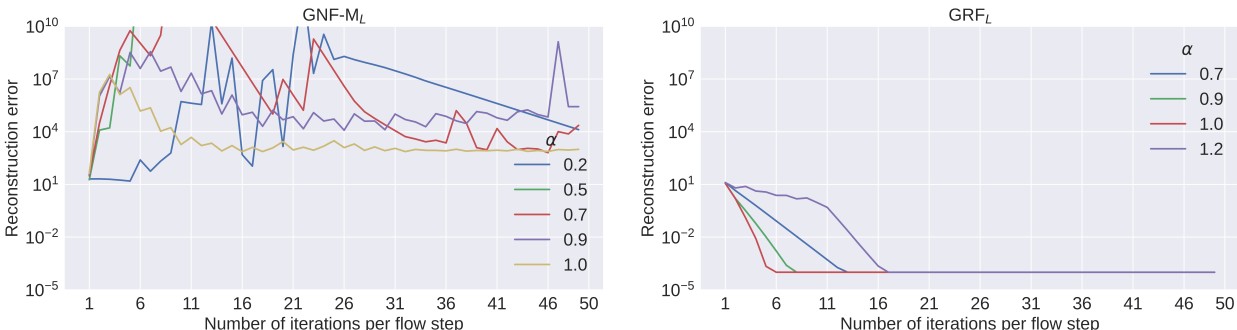

Figure 3: Reconstruction error when inverting GNF-M and GRF on 100 test samples from the Tree dataset. Note the large reconstruction error of GNF-M (left) for different values of the step-size, $\alpha$, compared to the fast convergence of GRF (right) to the correct inverse.

## 5.2 SIReN-VAE

As with GRFs, we evaluate the SIReN-VAE model on various synthetic and real-world datasets that each have an associated BN graph. For each BN, all leaf nodes are considered observed, and the rest are taken to be latent. Since VAEs are typically used to encode information into a lower-dimensional representation, we only consider the datasets for which there are fewer latent than observed variables, i.e. EColi and MEHRA. Additionally, we use an altered version of the Arithmetic Circuit dataset, where the number of observed variables has been increased from 2 to 10, as well as a larger linear Gaussian BN, Arth (Scutari, 2022). To better compare the effect of the encoded structure, we train three SIReN-VAE models that each incorporate a different structure. These are either fully-connected (SIReN-VAE$_{\text{FC}}$), random (SIReN-VAE$_{\text{Rand}}$), or adhere to the true dependencies (SIReN-VAE$_{\text{True}}$). For SIReN-VAE$_{\text{FC}}$, each observed variable is conditioned on all latent variables. SIReN-VAE$_{\text{Rand}}$ encodes a BN graph with the same number of edges as the graph encoded by SIReN-VAE$_{\text{True}}$, but where these edges have been assigned to random pairs of vertices. All models used the same latent dimension as SIReN-VAE$_{\text{True}}$. Details on the model architectures for each dataset is given in Appendix B.2. We also compare the results to those of a vanilla VAE, and a model (VAE+IAF+MAF) that is similar to SIReN-VAE$_{\text{FC}}$ except that the latent prior and approximate posterior were implemented with a masked autoregressive flow (MAF) (Papamakarios et al., 2017) and an inverse autoregressive flow (IAF) (Kingma et al., 2016), respectively. The decoder network of both these models, as well as the encoder network for the vanilla VAE, had similar architectures to the DecoderNN used in the SIReN-VAE models, i.e., the same activation functions, number of hidden layers and hidden layer widths were used. The MAF and IAF each consisted of three transformation steps, where the order of the input features was reversed in successive steps, as recommended by Kingma et al. (2016).

### 5.2.1 Performance in a Data-sparse Setting

We compared training the models on the full training sets of the respective datasets against using much smaller training sets consisting of only $2 \times |\mathcal{G}|$ instances, where $|\mathcal{G}| = D + K$. We noted each model's average negative log-evidence on the test set over five independent runs. The log-evidence per test point is estimated using 50 importance-weighted samples as in Burda et al. (2016). We also determined the average number of latent variables that collapsed during training. To measure whether a specific variable $z$ has collapsed, we use the statistic presented by Burda et al. (2016): $A_z = \text{Var}_{\mathbf{x} \sim p_D}(\mathbb{E}_{z \sim q(z|\mathbf{x})}[z])$. This is based on the assumption that if a latent dimension encodes useful information about the data, then its posterior mean would be expected to vary as the observations change. As in Burda et al. (2016), $z$ is deemed inactive if $A_z \leq 0.01$. These results are given in Table 4. In these initial experiments, SIReN-VAE$_{\text{True}}$ is not able to match the performance of SIReN-VAE$_{\text{FC}}$ in terms of log-evidence on most of the datasets when enough training data is available. Furthermore, using GRFs to construct the latent space for fully-connected SIReN-VAEs in this setting provided comparable performance to using affine autoregressive flows with feature shuffling between flow steps. In all cases, SIReN-VAE$_{\text{Rand}}$ did significantly worse. This shows that the encoded structure does play a significant role in modelling performance and that using the true (or hypothesised) BN structure aids in learning appropriate observational and latent distributions. When only limited training data is available, SIReN-VAE$_{\text{True}}$ clearly outperforms the other models, and achieves a higher log-evidence on most of the datasets. We speculate that the increased sparsity of the neural network weights, in line with the true BN independencies, poses an easier learning task and results in better generalization performance.

### 5.2.2 Addressing Posterior Collapse

As seen in Table 4, SIReN-VAE$_{\text{True}}$ suffers from posterior collapse. We are especially motivated in this setting to avoid this in order to learn meaningful latent distributions in line with the provided BN structure. We first investigate whether the position of a latent variable within the BN graph affects the likelihood that it will collapse, which would suggest that the encoded structure further contributes to inactive latents apart from known causes. We visualize in Figure 4 the logarithm of the average value of $A_z$ for each latent variable over the five runs used to generate the results in Table 4 for the two datasets in which posterior collapse occurred for the full training set. If the encoded structure plays no role, we expect the indices of collapsed variables to be arbitrary, resulting in all latent variables having a similar average activity score. This is what we would expect for the vanilla VAE, where all $z$ are independent and have the same prior distribution.

Table 4: Negative log-evidence ($-\log p(\mathbf{x})$) achieved by each model when trained on different sized training sets, as well as the number of collapsed latent variables. Each entry corresponds to the average over five runs with standard deviation given in the subscript. Bold indicates the best result in each group.

| | D | K | E | Model | $2 \times \|\mathcal{G}\|$ training samples | | All training samples | |
|---|---|---|---|---|---|---|---|---|
| | | | | | $-\log p(\mathbf{x})$ | #Inactive Units | $-\log p(\mathbf{x})$ | #Inactive Units |
| Arithmetic Circuit 2 | 10 | 5 | 15 | VAE | $13.90_{\pm 0.73}$ | $\mathbf{0.00_{\pm 0.00}}$ | $9.79_{\pm 0.01}$ | $2.00_{\pm 0.00}$ |
| | | | | VAE+IAF+MAF | $13.27_{\pm 0.75}$ | $\mathbf{0.00_{\pm 0.00}}$ | $\mathbf{9.76_{\pm 0.00}}$ | $\mathbf{0.00_{\pm 0.00}}$ |
| | | | | SIReN-VAE$_{\text{FC}}$ | $13.14_{\pm 0.78}$ | $\mathbf{0.00_{\pm 0.00}}$ | $\mathbf{9.76_{\pm 0.02}}$ | $\mathbf{0.00_{\pm 0.00}}$ |
| | | | | SIReN-VAE$_{\text{Rand}}$ | $14.86_{\pm 1.39}$ | $2.00_{\pm 0.89}$ | $11.09_{\pm 0.32}$ | $1.20_{\pm 0.75}$ |
| | | | | SIReN-VAE$_{\text{True}}$ | $\mathbf{12.29_{\pm 0.84}}$ | $2.60_{\pm 0.49}$ | $10.03_{\pm 0.01}$ | $2.00_{\pm 0.00}$ |
| EColi | 29 | 15 | 59 | VAE | $40.91_{\pm 1.07}$ | $\mathbf{0.00_{\pm 0.00}}$ | $35.02_{\pm 0.02}$ | $4.33_{\pm 1.25}$ |
| | | | | VAE+IAF+MAF | $39.57_{\pm 1.33}$ | $\mathbf{0.00_{\pm 0.00}}$ | $35.04_{\pm 0.01}$ | $\mathbf{0.00_{\pm 0.00}}$ |
| | | | | SIReN-VAE$_{\text{FC}}$ | $38.84_{\pm 0.22}$ | $\mathbf{0.00_{\pm 0.00}}$ | $35.02_{\pm 0.03}$ | $\mathbf{0.00_{\pm 0.00}}$ |
| | | | | SIReN-VAE$_{\text{Rand}}$ | $43.69_{\pm 0.40}$ | $2.00_{\pm 0.89}$ | $41.63_{\pm 0.53}$ | $1.80_{\pm 1.47}$ |
| | | | | SIReN-VAE$_{\text{True}}$ | $\mathbf{37.54_{\pm 0.33}}$ | $4.20_{\pm 0.75}$ | $\mathbf{34.99_{\pm 0.01}}$ | $\mathbf{0.00_{\pm 0.00}}$ |
| Arth | 67 | 40 | 150 | VAE | $64.22_{\pm 2.14}$ | $\mathbf{0.00_{\pm 0.00}}$ | $\mathbf{37.45_{\pm 0.03}}$ | $23.33_{\pm 6.34}$ |
| | | | | VAE+IAF+MAF | $42.79_{\pm 0.25}$ | $2.40_{\pm 1.62}$ | $37.59_{\pm 0.01}$ | $\mathbf{1.80_{\pm 1.17}}$ |
| | | | | SIReN-VAE$_{\text{FC}}$ | $43.35_{\pm 0.10}$ | $2.40_{\pm 1.36}$ | $37.56_{\pm 0.07}$ | $6.33_{\pm 1.89}$ |
| | | | | SIReN-VAE$_{\text{Rand}}$ | $52.74_{\pm 6.16}$ | $3.80_{\pm 3.19}$ | $40.81_{\pm 0.18}$ | $5.20_{\pm 2.04}$ |
| | | | | SIReN-VAE$_{\text{True}}$ | $\mathbf{41.56_{\pm 0.53}}$ | $19.40_{\pm 2.06}$ | $37.73_{\pm 0.05}$ | $14.80_{\pm 0.33}$ |
| MEHRA | 7 | 3 | 10 | VAE | $10.57_{\pm 0.21}$ | $\mathbf{0.00_{\pm 0.00}}$ | $7.65_{\pm 0.02}$ | $\mathbf{0.00_{\pm 0.00}}$ |
| | | | | VAE+IAF+MAF | $\mathbf{10.42_{\pm 0.33}}$ | $\mathbf{0.00_{\pm 0.00}}$ | $7.61_{\pm 0.02}$ | $\mathbf{0.00_{\pm 0.00}}$ |
| | | | | SIReN-VAE$_{\text{FC}}$ | $10.56_{\pm 0.32}$ | $0.50_{\pm 0.50}$ | $\mathbf{7.58_{\pm 0.01}}$ | $\mathbf{0.00_{\pm 0.00}}$ |
| | | | | SIReN-VAE$_{\text{Rand}}$ | $10.65_{\pm 0.04}$ | $1.80_{\pm 0.40}$ | $8.92_{\pm 0.23}$ | $\mathbf{0.00_{\pm 0.00}}$ |
| | | | | SIReN-VAE$_{\text{True}}$ | $10.60_{\pm 0.05}$ | $1.33_{\pm 1.11}$ | $8.37_{\pm 0.06}$ | $\mathbf{0.00_{\pm 0.00}}$ |

Considering Figure 4, this is indeed the case as the average activity of the vanilla VAE's latent variables is far more uniform (compared to SIReN-VAE$_{\text{True}}$). Interestingly, for SIReN-VAE$_{\text{FC}}$, the higher the index of the latent, the lower its activity on average. For the fully-connected BNs, each variable is conditioned on all variables with a lower index. This suggests that the model may tend to de-emphasize latent variables that are conditioned on many others. Figure 4 shows that in SIReN-VAE$_{\text{True}}$, posterior collapse tends to happen for certain variables far more than for others. Further inspecting the positions of the collapsed $z$ in the BNs shows that the corresponding vertices typically only share edges with other latent variables, and not with any observed variables. This aligns with the findings of Burda et al. (2016), who observed that the latents in higher layers of a hierarchical VAE (which do not directly influence any $\mathbf{x}$) are more prone to collapse.

Since we would like the learned latent distribution of SIReN-VAE$_{\text{True}}$ to respect the provided BN structure and utilize all the latent dimensions, the fact that aspects of the encoded structure make posterior collapse more likely, makes it doubly important for us to try to address it in some way. We therefore investigate the efficacy of warm-up (WU) (Bowman et al., 2016) and importance-weighted (IW) (Burda et al., 2016) objectives in combatting posterior collapse, with the results given in Table 5.[5] Since we have found that a certain subset of latent variables is more prone to collapse than others, we compare two different approaches to warm-up. The standard, more naïve, approach applies the warm-up factor to all latent variables, and is denoted by WU$_{\text{all}}$. We also apply the warm-up term to only those latent variables that typically collapsed, as identified in Figure 4. We denote this 'retrained' approach by WU$_{\text{select}}$. We do not apply warm-up if no posterior collapse occurred on the full training set as indicated in Table 4. For further information, see Appendix B.2. For comparison, we also apply these techniques to SIReN-VAE$_{\text{FC}}$.

---

[5]We also considered using lower-variance gradient estimates (Tucker et al., 2018) to combat posterior collapse. Although this approach generally provided slight performance gains, it comes at a significant time and memory cost—see Appendix B.3.4.

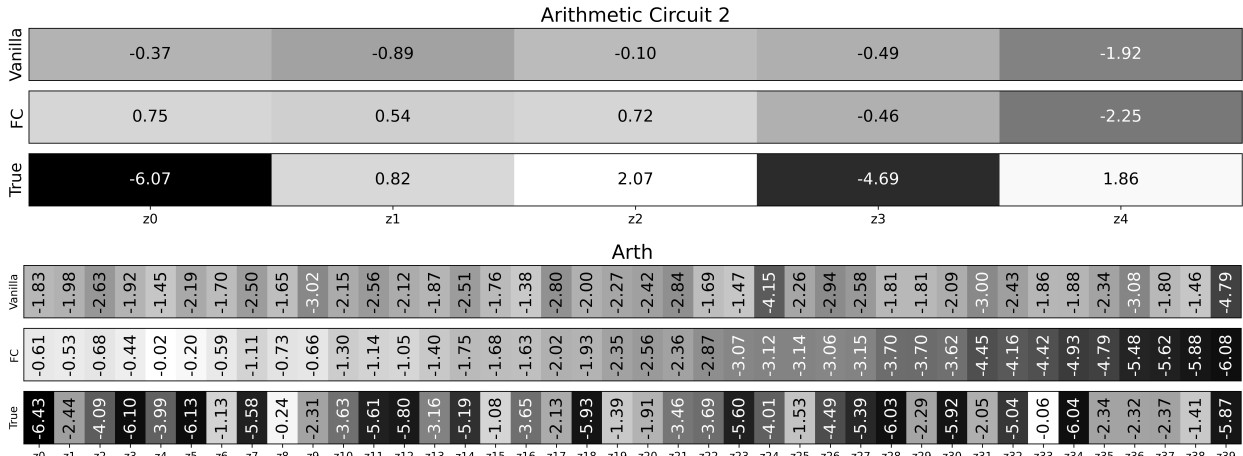

Figure 4: The (logarithm of the) average of the posterior collapse statistic $A_{z_i}$, for each of the latent variables of the Arithmetic Circuit 2 and Arth datasets. The top, middle and bottom bar for each dataset corresponds to the vanilla VAE, SIReN-VAE$_{\text{FC}}$ and SIReN-VAE$_{\text{True}}$, respectively. The lower this value, the darker the corresponding block. Note that $\log(0.01) \approx -4.6$ is the point at which a latent variable is deemed collapsed.

Table 5: Performance of SIReN-VAE when applying posterior collapse mitigation techniques, namely: warm-up (WU) and importance weighted objective (IWAE). Warm-up is applied to either all latents (WU$_{\text{all}}$), or to a selected subset that is prone to collapse (WU$_{\text{select}}$).

| | | $2 \times |\mathcal{G}|$ training samples | | All training samples | |
|---|---|---|---|---|---|
| | Model | $-\log p(\mathbf{x})$ | #Inactive Units | $-\log p(\mathbf{x})$ | #Inactive Units |
| Arithmetic Circuit 2 | SIReN-IWAE$_{\text{FC}}$ | $12.78_{\pm 0.43}$ | $\mathbf{0.00}_{\pm 0,00}$ | $\mathbf{9.73}_{\pm 0.01}$ | $\mathbf{0.00}_{\pm 0,00}$ |
| | SIReN-IWAE$_{\text{True}}$+WU$_{\text{all}}$ | $\mathbf{11.88}_{\pm 0.27}$ | $\mathbf{0.00}_{\pm 0,00}$ | $9.86_{\pm 0.08}$ | $1.00_{\pm 0.00}$ |
| | SIReN-IWAE$_{\text{True}}$+WU$_{\text{select}}$ | $11.96_{\pm 0.40}$ | $\mathbf{0.00}_{\pm 0,00}$ | $9.80_{\pm 0.00}$ | $\mathbf{0.00}_{\pm 0,00}$ |
| EColi | SIReN-IWAE$_{\text{FC}}$ | $38.86_{\pm 0.28}$ | $\mathbf{0.00}_{\pm 0,00}$ | $35.04_{\pm 0.01}$ | $\mathbf{0.00}_{\pm 0,00}$ |
| | SIReN-IWAE$_{\text{True}}$ | $\mathbf{38.42}_{\pm 0.98}$ | $1.60_{\pm 0.80}$ | $\mathbf{34.98}_{\pm 0.00}$ | $\mathbf{0.00}_{\pm 0,00}$ |
| Arth | SIReN-IWAE$_{\text{FC}}$+WU$_{\text{all}}$ | $43.02_{\pm 0.80}$ | $\mathbf{7.40}_{\pm 3.93}$ | $\mathbf{37.32}_{\pm 0.18}$ | $1.00_{\pm 0.00}$ |
| | SIReN-IWAE$_{\text{True}}$+WU$_{\text{all}}$ | $\mathbf{41.09}_{\pm 0.42}$ | $15.00_{\pm 1.26}$ | $37.48_{\pm 0.47}$ | $10.20_{\pm 1.33}$ |
| | SIReN-IWAE$_{\text{True}}$+WU$_{\text{select}}$ | $41.31_{\pm 0.43}$ | $17.60_{\pm 1.85}$ | $37.42_{\pm 0.24}$ | $7.00_{\pm 2.28}$ |
| MEHRA | SIReN-IWAE$_{\text{FC}}$ | $10.47_{\pm 0.45}$ | $\mathbf{0.33}_{\pm 0.75}$ | $\mathbf{7.58}_{\pm 0.02}$ | $\mathbf{0.00}_{\pm 0,00}$ |
| | SIReN-IWAE$_{\text{True}}$ | $\mathbf{10.36}_{\pm 0.27}$ | $0.50_{\pm 0.50}$ | $8.19_{\pm 0.06}$ | $\mathbf{0.00}_{\pm 0,00}$ |

Table 5 shows that these techniques help to improve the performance of SIReN-VAE$_{\text{True}}$ and reduce the number of inactive latent variables. However, when enough data is available, SIReN-VAE$_{\text{True}}$ still does not quite match the performance of SIReN-VAE$_{\text{FC}}$, especially on the real-world MEHRA dataset. This could be because BNs constructed for real-world domains tend to focus more on the key influences between variables and neglect smaller interactions, which could limit SIReN-VAE$_{\text{True}}$'s ability to achieve comparable log-evidence scores. Promisingly, SIReN-VAE$_{\text{True}}$ performed the best on MEHRA in the data-sparse setting when employing an IW objective. It is perhaps precisely the coarser-grained structure encoded by SIReN-VAE$_{\text{True}}$ that prevents it from overfitting and allows better generalization performance.

### 5.2.3 Interpretability of the Latent Space

One of the anticipated benefits of incorporating hypothesized BN structures into a VAE is a more interpretable latent space. Since we have access to the true latent variables of the datasets,[6] we investigated whether this is being realized by estimating the mutual information (MI) between each component of the true latent, $\mathbf{z}^*$, and each component of the latent inferred by the model, $\mathbf{z}$, denoted by $\mathrm{I}(z_i^*, z_j)$ for $i, j = 1, \ldots, K$, using the MI neural estimator (MINE) of Belghazi et al. (2018). The samples used to estimate MI were obtained by sampling a single point $\mathbf{z}_n$ from $q_\phi(\cdot|\mathbf{x}_n)$ for each pair $(\mathbf{x}_n, \mathbf{z}_n^*)$ in the training set. Figure 5 gives these results. We used the SIReN-VAE$_{\text{True}}$ models that achieved the best log-evidence as given in Table 5 above. In Figure 6, we also plot the MI between pairs of true latent and observed variables, and pairs of latent and observed variables sampled from SIReN-VAE$_{\text{True}}$ for the EColi dataset.

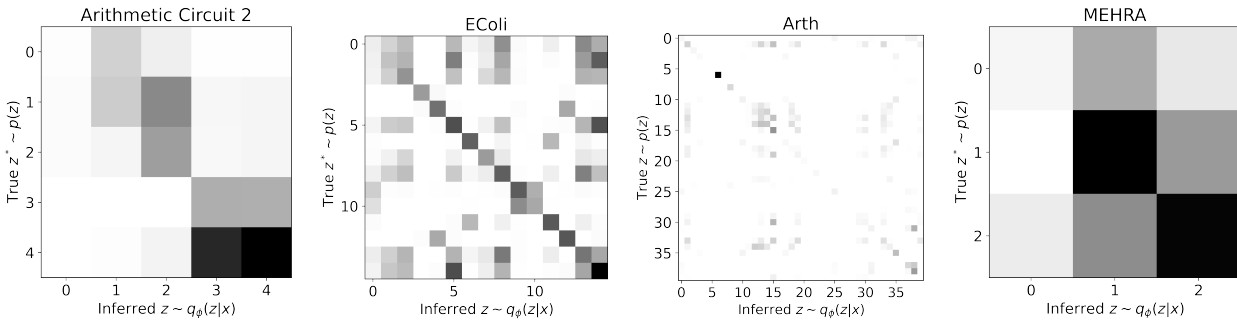

Figure 5: Visualization of the MI between latent variables. Each entry corresponds to MI between the true and inferred latent variables, $I(z_i^*, z_j)$ for $i, j = 1, \ldots, K$, computed using the MI neural estimator (MINE) of Belghazi et al. (2018). The higher the MI, the darker the associated entry.

If SIReN-VAE$_{\text{True}}$ learns meaningful latent representations, with each $z_i$ affecting its neighbourhood in a way similar to the true hidden variable, we would expect the MI between $z_i^*$ and $z_i$ to be high and for $\mathrm{I}(z_i^*, z_j) \approx \mathrm{I}(z_j^*, z_i)$. Visually, this means that we expect the matrix plots of Figure 5 to be approximately symmetric and to have a noticeable line on the diagonal. This holds the most noticeably for the EColi dataset. Although arguably not as distinct, similar trends are evident for the other datasets as well, e.g. for the real-world MEHRA dataset, two out of the three latent variables of the model have very high MI with their true counterparts. The two plots in Figure 6 are also nearly identical, meaning that SIReN-VAE$_{\text{True}}$ closely mimics the dependencies of the true model in terms of the MI between variables. The results support our hypothesis that SIReN-VAE$_{\text{True}}$ should provide better interpretability in that one gains further insight into which latent variables directly influence each other. This could aid in more controlled conditional sample generation, and could allow one

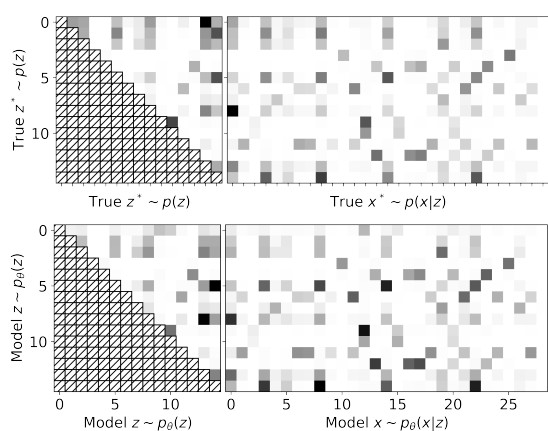

Figure 6: MI between latent and observed variables for the EColi dataset.

to associate specific roles with each latent factor of SIReN-VAE$_{\text{True}}$, corresponding to their meaning in the original BN, as for example in the MEHRA BN where the latent variables represent atmospheric conditions.

---

[6]For the synthetic datasets, the latent variables were sampled from the known models. For the real-world MEHRA dataset, we assumed certain dimensions to be latent for our purposes, and as such have access to the true 'latent' data.

## 6 Discussion

**Limitations & Future Work** BNs are commonly defined over sets of both continuous and discrete variables. Extending GRFs and SIReN-VAEs to support discrete variables would thus greatly broaden their applicability. One desirable direction for future work is therefore to investigate integrating gradient estimation approaches for discrete variables (e.g. Bengio et al. (2013); Tran et al. (2019)) into these models.

In this work, we assumed access to an appropriate BN describing the dependency structure between the variables of interest. However, in most real-world settings one does not typically have access to such a structure, or one only has partial knowledge about the dependency graph. Therefore it is of interest to learn or refine the dependency structure from data. This idea has been explored in the context of normalizing flows and VAEs (e.g. (Wehenkel & Louppe, 2021; He et al., 2019)), and is an interesting future line of work to incorporate into the GRF and SIReN-VAE models.

VAEs are typically applied to high-dimensional data like images and audio. As mentioned above, suitable BN structures are usually not readily available in these settings. In certain contexts, one might posit an appropriate latent structure, but it is not clear how to associate specific portions of the observation with specific latents in a more fine-grained way than using full connectivity between certain latents and all input features. Initial experiments using SIReN-VAE (with a convolutional decoder) on synthetically generated images using a known sparse latent dependency structure and full connectivity between latents and observed features did not provide interpretable latent variables. Further exploration is therefore required to determine how best to apply SIReN-VAE directly to such high-dimensional data, and what a sensible way might be to identify meaningful finer dependency structures between latent variables and observations.

**Conclusion** We proposed the GRF as an alternative to existing graphical NFs such as GNF-M (Wehenkel & Louppe, 2021) and SCCNF (Weilbach et al., 2020). While these flows provide very good modelling capability, they do not ensure stable inversion. GRFs on the other hand, exhibited comparable modelling performance, while being designed for reliable and more time-efficient inversion. The GRF is thus a suitable option to use when some assumed dependency structure is available and where the flow may be required to perform reliably in both directions. This is for example needed in the prior of our proposed SIReN-VAE model, which is an approach for incorporating prior information from an arbitrary BN graph into a VAE. Encoding this structure does not lead to significantly better generalization performance than simply using a fully-connected structure when enough data is available. The key benefits of SIReN-VAE$_{\text{True}}$ are its ability to provide more nuanced interpretability and to allow stable training with good generalization on small training sets. Practitioners who elicit BNs typically omit unknown factors, because they cannot in general use them in current modelling frameworks. Based on our results, we suggest that SIReN-VAE$_{\text{True}}$ might be of use when practitioners know or can hypothesize about the dependency structure of certain latent factors in their domain, especially if they wish to interpret the model according to the chosen dependency structure. SIReN-VAE$_{\text{True}}$ could also facilitate higher quality data augmentation from limited real-world data.

## 7 Acknowledgements

We would like to thank DeepMind for financially supporting J. Mouton through a scholarship, as well as the reviewers for helping fine-tune the ideas in this paper.

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

# A  Appendix: Additional Theoretical Background

## A.1  Extending MADE for Arbitrary Graphical Structures

Suppose a joint distribution factorizes as

$$p(\mathbf{x}) = \prod_{i=1}^{D} p(x_i | \text{Pa}_{x_i}^{\mathcal{G}}) \tag{8}$$

for DAG $\mathcal{G}$. $\text{Pa}_{x_i}^{\mathcal{G}}$ denotes the parents of $x_i$ in $\mathcal{G}$. For a given neural network that takes $\mathbf{x}$ as input, the goal is to have the output units associated with $x_i$ be computed from only those input units associated with $x_i$ and its parents. This means that there should be no computational paths between an input and an output unit if there is no direct dependency between the associated variables in $\mathcal{G}$. This can be achieved by applying a masking matrix to the weights of each neural network layer (which can be of arbitrary width) such that at least one weight on any such computational path is set to zero.

We follow a similar approach to MADE (Germain et al., 2015), which constructs masks for an implicit fully-connected BN. We begin by assigning a specific subset of variables to each unit in the neural network. Specifically, each input unit is assigned a unit set containing its corresponding input variable: $\{x_i\}$. Each output unit is assigned a set consisting of its associated variable and that variable's parents in the BN: $\{x_i\} \cup \text{Pa}_{x_i}^{\mathcal{G}}$. Lastly, each hidden unit is randomly assigned one of the following sets: $\{x_i\}$ or $\{x_i\} \cup \text{Pa}_{x_i}^{\mathcal{G}}$ where $i$ can be any of $1, \ldots, D$.[7]

A correct mask can then be constructed by ensuring that it zeroes out any weight between two neural network units if the set assigned to the unit in the next layer is not a superset of the set assigned to the unit in the previous layer. This has the implication that any path from input to output for any variable has a single associated set switch from $\{x_i\}$ to $\{x_i\} \cup \text{Pa}_{x_i}^{\mathcal{G}}$.

## A.2  Invertibility of Graphical Normalizing Flows in Practice

The Lipschitz constants of the forward and inverse transformation of an invertible neural network quantifies its worst-case stability. Bounds on these values play an important role in understanding and mitigating possible exploding inverses.

**GNF-A**  No global bounds can be placed on the Lipschitz constant of this type of flow, which complicates the task of ensuring stable inversion in all scenarios. Behrmann et al. (2021) provide the following simple illustration of why GNF-A only has local Lipschitz bounds. Assume $\mathbf{x}$ consists of two variables, $x_0$ and $x_1$, where $x_1$ is dependent on $x_0$ in the corresponding BN. Let $[F(\mathbf{x})]_1 = x_1 \exp(s(x_0))$ be the transformation applied to $x_1$ by a single-step GNF-A, where $s(x_0) = [s(\mathbf{x})]_1$ is the second output dimension of the conditioner function $s(\cdot)$ and the dependence on only $x_0$ has been made explicit. The output of the conditioner function $[m(\mathbf{x})]_1$ is taken to be 0 for simplicity. Then

$$\frac{\partial [F(\mathbf{x})]_1}{\partial x_0} = x_1 \frac{\partial \exp(s(x_0))}{\partial x_0} = x_1 \exp(s(x_0)) s'(x_0) . \tag{9}$$

Thus, if $x_1$ may grow arbitrarily large, this derivative will be unbounded, which could allow the Jacobian, $J_F(\mathbf{x})$, to have an unbounded Frobenius norm. Due to the equivalence of norms in finite dimensions, this in turn can induce an unbounded spectral norm of the Jacobian. Lastly, we consider the following theorem from Federer (1996): if $F : \mathbb{R}^D \to \mathbb{R}^D$ is a Lipschitz continuous and differentiable function under the Euclidean norm, then

$$\text{Lip}(F) = \sup_{\mathbf{x} \in \mathbb{R}^D} ||J_F(\mathbf{x})||_2, \tag{10}$$

where $|| \cdot ||_2$ denotes the spectral norm. Based on Equation (10), we conclude that if the spectral norm of the Jacobian is unbounded, then no global Lipschitz bound can be obtained.

---

[7]To prevent situations where there are no valid paths from an input to the corresponding output, we also require at least one unit in each hidden layer associated with $\{x_i\}$.

**GNF-M**   We can employ a similar illustration to investigate the Lipschitz bounds of a GNF with monotonic transformations. Again, assume $\mathbf{x}$ consists of two variables, $x_0$ and $x_1$, where $x_1$ depends on $x_0$ in the corresponding BN. The transformation applied to $x_1$ by a single-step GNF-M is then given by $[F(\mathbf{x})]_1 = \int_0^{x_1} h(t, c_1(x_0)) \, dt + \beta(c_1(x_0))$. We take the partial derivative of the above transformation and apply Leibniz's integral rule and the chain rule:

$$
\begin{aligned}
\frac{\partial [F(\mathbf{x})]_1}{\partial x_0} &= \frac{\partial}{\partial x_0} \int_0^{x_1} h(t, c_1(x_0)) \, dt + \frac{\partial \beta(c_1(x_0))}{\partial x_0} \\
&= \int_0^{x_1} \frac{\partial h(t, c_1(x_0))}{\partial c_1(x_0)} \frac{\partial c_1(x_0)}{\partial x_0} \, dt + \frac{\partial \beta(c_1(x_0))}{\partial x_0} \quad .
\end{aligned}
\tag{11}
$$

The integrand above is the product of the derivatives of two neural networks with respect to their inputs. For general networks, this integrand's shape will depend not only on the chosen activation functions, but also the weights obtained during training. If $x_1$ may grow arbitrarily large, and if the area under the curve given by the integrand is not bounded by some maximum value as $x_1$ increases, then we can apply similar reasoning as above to show that this flow has no global Lipschitz bounds. Thus, either the architecture of the flow must be adapted to ensure that this integral remains bounded as a function of $x_1 > 0$, or other techniques must be used to improve local stability, as discussed in Behrmann et al. (2021).

**SCCNF**   Given that the flow is defined by a neural ODE, $\frac{d\mathbf{x}(t)}{dt} = f(\mathbf{x}(t), t)$, where $t \in [0, 1]$, we have that the Lipschitz constant for both the forward and inverse transformation are upper bounded by $e^{\mathrm{Lip}(f) \cdot t}$ (Behrmann et al., 2021).

### A.3   Dependence Structure Induced by GRFs

We investigate whether the distribution represented by a given GRF does indeed encode all the conditional independencies specified by the provided BN. We first consider a normalizing GRF with a single flow step, $F(\mathbf{x}) = \boldsymbol{\epsilon}$, which encodes the following BN chain structure: $x_0 \to x_1 \to x_2$. Let the bijective transformations applied to each of the dimensions be given by $F_0(x_0) = \epsilon_0$, $F_1(x_1; x_0) = \epsilon_1$ and $F_2(x_2; x_1) = \epsilon_2$. A graphical illustration of this flow is given in Figure 7a. Note that these individual bijective transformations are implemented using a single residual block neural network. We treat them separately here to simplify the discussion. We are interested in whether the distribution represented by the flow, $p(\mathbf{x})$, respects the conditional independence assumptions specified by this BN.

After a single transformation step, the distribution of $x_2$ will only depend on $x_1$ as desired, since it can be computed as: $\log p(x_2|x_1) = \log p_0(F_2(x_2; x_1)) + \log |\det(J_{F_2}(x_2; x_1))|$. This illustrates how the distribution represented by a normalizing GRF with a single transformation step will adhere to the conditional independencies specified by the BN, which is in line with the argument presented by Wehenkel & Louppe (2021).

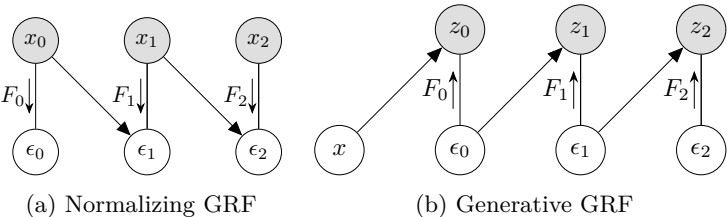

(a) Normalizing GRF          (b) Generative GRF

Figure 7: A graphical illustration of the transformation applied by (a) a one-step normalizing GRF and (b) a one-step generative GRF when encoding the chain dependency structures: $x_0 \to x_1 \to x_2$ and $x \to z_0 \to z_1 \to z_2$, respectively. The variables of the base distribution are represented by $\epsilon_0$, $\epsilon_1$ and $\epsilon_2$. Undirected edges represent a bijective transformation ($F_0$, $F_1$ or $F_2$) between the associated variables with the small arrow indicating the direction of the forward mapping of the flow. Directed edges indicate the additional variables these bijective transformations are conditioned on as enforced by the presented masking scheme.

Additional dependencies are however introduced when the number of flow steps is increased. Considering Figure 8, which depicts a 2-step normalizing GRF, one can note that when computing the latent representation of $x_2$, information will 'leak' from $x_0$ via the intermediate transformations of the observed variables. If enough transformation steps are applied, the distribution of any observed variable will ultimately depend on all its ancestors in the BN graph. As a result, the encoded structure may end up corresponding to the transitive closure of the original BN structure.

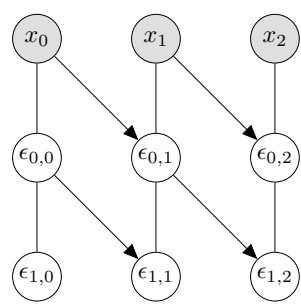

Figure 8: A normalizing GRF with two transformation steps.

Next, consider a *generative* GRF with a single flow step, $\mathbf{z} = F(\boldsymbol{\epsilon}; x)$ for $\boldsymbol{\epsilon} \sim p_0$, which encodes the following BN chain structure: $x \to z_0 \to z_1 \to z_2$. Again, let the bijective transformations applied to each of the dimensions be given by $z_0 = F_0(\epsilon_0; x)$, $z_1 = F_1(\epsilon_1; \epsilon_0)$ and $z_2 = F_2(\epsilon_2; \epsilon_1)$, where each transformation is conditioned on a subset of $\boldsymbol{\epsilon}$, and only $F_0$ is conditioned on the observation, $x$. A graphical illustration of this flow is given in Figure 7b. We are interested in whether the distribution of the generated samples, $q(\mathbf{z}|x)$, respects the conditional independence assumptions specified by this BN, such as that $z_2$ is conditionally independent of $z_0$ and $x$, given $z_1$. That is, $q(z_2|z_0, z_1, x) = q(z_2|z_1)$ should hold in the distribution represented by the flow. We can compute:

$$
\begin{aligned}
\log q(z_2|z_0, z_1, x) &= \log p_0(F_2^{-1}(z_2; \epsilon_1)) + \log \left| \det(J_{F_2^{-1}}(z_2; \epsilon_1)) \right| \\
&= \log p_0(F_2^{-1}(z_2 | F_1^{-1}(z_1; \epsilon_0))) + \log \left| \det(J_{F_2^{-1}}(z_2; F_1^{-1}(z_1; \epsilon_0))) \right| \\
&= \log p_0(F_2^{-1}(z_2; F_1^{-1}(z_1; F_0^{-1}(z_0; x)))) + \log \left| \det(J_{F_2^{-1}}(z_2; F_1^{-1}(z_1; F_0^{-1}(z_0; x)))) \right| .
\end{aligned}
$$

By expanding the expression in this way, we make clear the direct dependence of $z_2$ on $z_0$ and $x$—knowing only $z_1$ is not sufficient to specify $q(z_2|z_1, z_0, x)$. This dependence arises from the fact that the bijective transformation between $z_2$ and $\epsilon_2$ is only specified once $\epsilon_1$ is known, and $\epsilon_1$ is a function of both $z_1$ *and* $z_0$. Thus, $q(z_2|z_0, z_1, x) \neq q(z_2|z_1)$. In this way, each variable could be dependent on all its ancestors in the BN, and the dependency structure induced by the flow may again ultimately correspond to the transitive closure of the encoded BN.

We therefore have that for both a normalizing and generative GRF, the dependency structure induced by the flow could correspond to the encoded BN's transitive closure. In the *worst-case* scenario, this transitive closure corresponds to a fully-connected graph, in which case one would seemingly not have gained any benefit from encoding the given structure in this way. The dependencies induced by these graphical flows are arguably more subtle, however. For normalizing GRFs, each variable only receives information from its ancestors via intermediate bijective transformations of its parents. Even though there is some 'information leakage', it would not be unreasonable to expect that the distribution of a given variable will be more strongly influenced by its parents, rather than by the potentially 'diluted' information the variable receives about the rest of its ancestors. The degree of information leakage for generative GRFs is even less clear. This is primarily because each variable is in fact never a direct function of its parents, but rather depends on bijective transformations of these parents where each bijection is itself conditioned on those variables' parents. For example, we showed that knowing only the parent of $z_2$ is not sufficient to calculate its density (if $\boldsymbol{\epsilon}$ is unknown), and one additionally needs knowledge of its ancestors, including the observation $x$. When performing density estimation with a generative flow, this dependence is thus already introduced after only one flow step. When generating new samples, one however still needs three flow transformations for a generated sample of $z_2$ to contain *any* information about the observed state (similar to how $\epsilon_{1,2}$ only receives information about $x_0$ after two steps in the normalizing GRF depicted in Figure 8). Taken together, even though GRFs cannot in general guarantee that the distribution represented by the flow respects the independence statements specified by the encoded BN, we still expect these flows to incorporate a strong inductive bias that encourages the variables to adhere to the desired dependency structure (see Appendix B.3.3).

# B  Appendix: Datasets & Experiments

## B.1  Bayesian Network Datasets

**Arithmetic Circuit**   The synthetic arithmetic circuit BN is the same as used that by Weilbach et al. (2020) and Wehenkel & Louppe (2021). For density estimation tasks, all variables are observed. For amortized inference tasks, variables $z_0$ to $z_5$ are latent, while $x_0$ and $x_1$ are observed. This distribution consists of heavy-tailed densities which are linked through non-linear dependencies.

$$z_0 \sim \text{Laplace}(5, 1)$$
$$z_1 \sim \text{Laplace}(-2, 1)$$
$$z_2 \sim \mathcal{N}(\tanh(z_0 + z_1 - 2.8), 0.1)$$
$$z_3 \sim \mathcal{N}(z_0 \times z_1, 0.1)$$
$$z_4 \sim \mathcal{N}(7, 2)$$
$$z_5 \sim \mathcal{N}(\tanh(z_3 + z_4), 0.1)$$
$$x_0 \sim \mathcal{N}(z_3, 0.1)$$
$$x_1 \sim \mathcal{N}(z_5, 0.1).$$

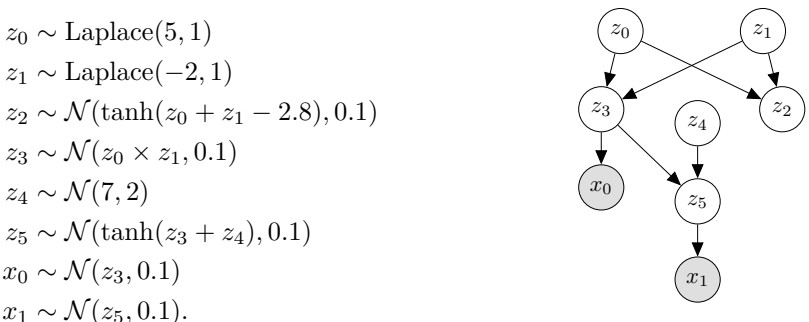

Figure 9: Arithmetic Circuit BN.

**Tree**   This is another synthetic dataset. It is adapted from the model given in Wehenkel & Louppe (2021) to obtain a fully-specified model for which the joint density can be computed in closed form (as needed for the inference tasks)—see Figure 10. Instead of using the circles 2D dataset from Grathwohl et al. (2019) as in Wehenkel & Louppe (2021), the first two variables are sampled from a 2D Gaussian mixture model, $\text{GMM}_2$, which consists of two equally weighted components with means at $(1, 1)$ and $(-1, -1)$ and shared covariance matrix $0.2 \times I_2$. As in Wehenkel & Louppe (2021), the second pair of variables is sampled from a GMM with 8 equally weighted components with means at $(0, 1.5)$, $(1, 1)$, $(1.5, 0)$, $(1, -1)$, $(0, -1.5)$, $(-1, -1)$, $(-1.5, 0)$ and $(-1, 1)$ and shared covariance matrix $0.1 \times I_8$.

$$z_0, z_1 \sim \text{GMM}_2$$
$$z_2, z_3 \sim \text{GMM}_8$$
$$z_4 \sim \mathcal{N}(\max(z_0, z_1), 1)$$
$$z_5 \sim \mathcal{N}(\min(z_2, z_3), 1)$$
$$x_0 \sim \mathcal{N}\left(\frac{1}{2}(\sin(z_4 + z_5) + \cos(z_4 + z_5)), 1\right)$$

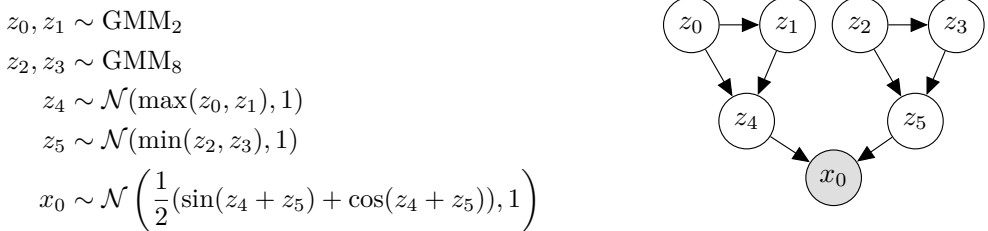

Figure 10: Tree BN.

**EColi**   The EColi dataset was generated using a fully-specified BN adapted from the BN repository of Scutari (2022). All vertices are (conditionally) Gaussian, with means given by a linear combination of the parents. Certain leaf vertices were removed from the original BN (`nmpc` and `ftsJ`) in order to obtain a BN with fewer latent than observed variables, making thr BN more applicable for the SIReN-VAE setting as well. The following vertices were considered to be latent: {`asnA, atpD, b1191, cspA, cspG, dnaK, eutG, fixC, icdA, lacA, lacY, sucA, yedE, ygcE, yheI`}, while the rest were considered observed. See Figure 11a for a diagram of the BN.

**Protein**   This real-world dataset consists of 11 observed variables containing information about multiple phosphorylated human proteins (Sachs et al., 2005)—see Figure 11b for a diagram of the BN structure. The BN structure encodes the cellular signalling network then believed to exist between these proteins.

**MEHRA** The Multi-dimensional Environment-Health Risk Analysis dataset (MEHRA) was assembled by Vitolo et al. (2018) to help model air pollution, climate and health in English regions using a BN. We only consider the subgraph of the BN obtained during the study, corresponding to the continuous variables, see Figure 11c. As such, only a subset of the original dataset corresponding to a fixed set of discrete variables is used. This reduced dataset consisted of all observations of the continuous variables for the following setting of the observed variables: {Region=Greater London Authority; Zone=Greater London Urban Area; Type=Traffic Urban; Year=2014; Season=Winter}.

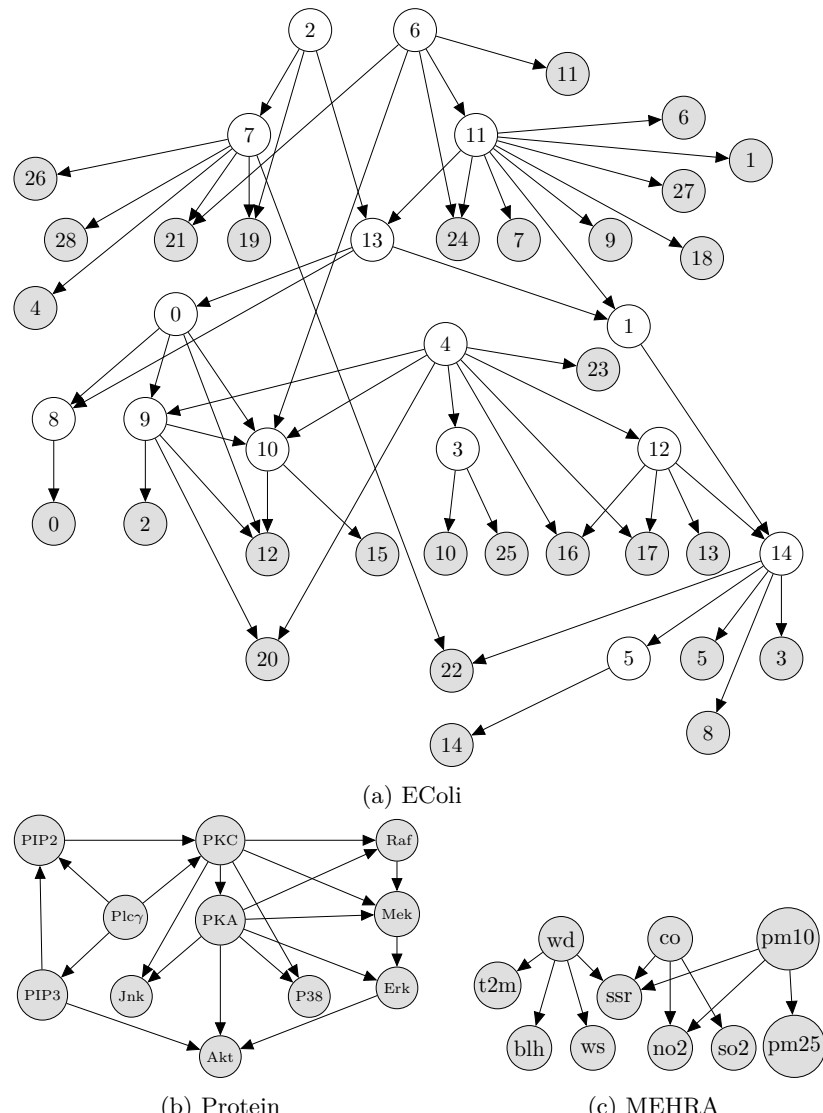

(a) EColi

(b) Protein

(c) MEHRA

Figure 11: BN graphs associated with the EColi, Protein and MEHRA datasets.

Since VAEs are typically used to encode information into a lower-dimensional representation, we only consider the datasets from the GRF investigation for which there are fewer latent than observed variables. Thus, in addition to EColi and MEHRA presented above we also use the following datasets:

**Arth** The Arth dataset was generated using a fully-specified BN from the BN repository of Scutari (2022). All vertices are (conditionally) Gaussian, with means given by a linear combination of the parents. All leaf vertices were considered to be observed, and the rest are latent. See Figure 12 for the corresponding BN graph.

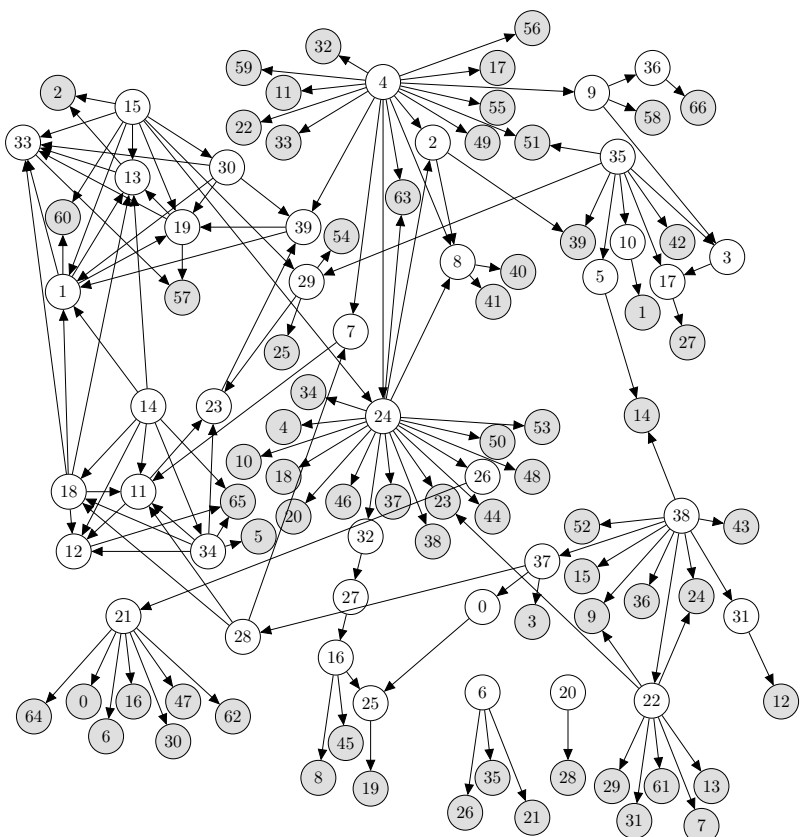

Figure 12: Arth BN graph.

**Arithmetic Circuit 2** This Arithmetic Circuit BN is an adaptation of the one described above in Figure 9, where we have added more observed variables. Variables $z_0$ to $z_4$ are latent, while $x_0$ to $x_9$ are observed. See Figure 13. This distribution consists of heavy-tailed densities which are linked through non-linear dependencies.

$$z_0 \sim \text{Laplace}(5,1)$$
$$z_1 \sim \text{Laplace}(-2,1)$$
$$z_2 \sim \mathcal{N}((z_0 \times z_1)/7.9 - 7, 0.1)$$
$$z_3 \sim \mathcal{N}(7,2)$$
$$z_4 \sim \mathcal{N}(\tanh(z_2 + z_3), 0.1)$$

$$x_0 \sim \mathcal{N}(\tanh(z_0 + z_1 - 2.8), 0.1)$$
$$x_1 \sim \mathcal{N}(\tanh(z_1), 1.1)$$
$$x_2 \sim \mathcal{N}(\tanh(z_2 + z_3), 0.1)$$
$$x_3 \sim \mathcal{N}(z_2 + 8, 0.1)$$
$$x_4 \sim \mathcal{N}(\sigma(z_3 - 7), 1.1)$$
$$x_5 \sim \mathcal{N}((z_2 \times z_4)/6.1, 0.1)$$
$$x_6 \sim \mathcal{N}(z_4, 1.1)$$
$$x_7 \sim \mathcal{N}(z_4, 0.1)$$
$$x_8 \sim \mathcal{N}(\tanh(z_4), 2.1)$$
$$x_9 \sim \mathcal{N}(\sin(z_4), 1.1).$$

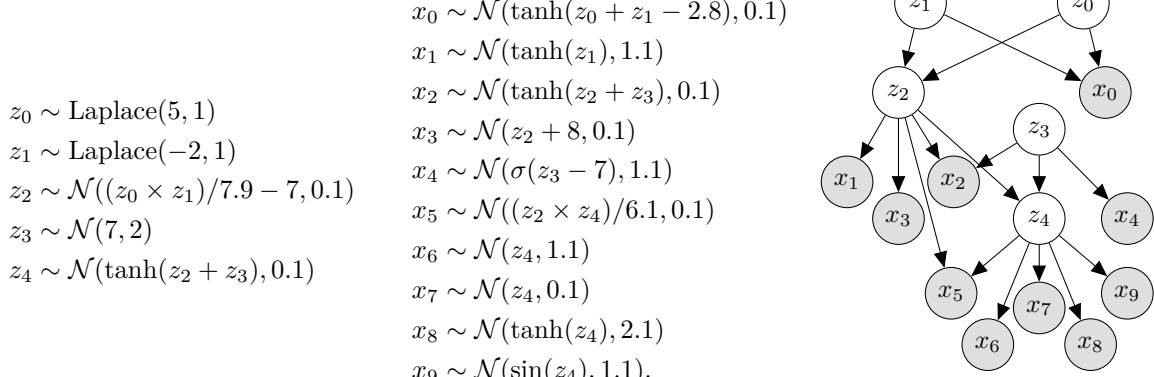

Figure 13: Arithmetic Circuit 2 BN.

## B.2 Model Architectures & Experiment Setup

**GRF** We train two models per task for each of the approaches. The first is a smaller model with a maximum capacity of 5000 trainable parameters, denoted by a subscript S. The second, larger model has a maximum capacity of 15000 parameters, denoted by the subscript L. Table 6 details the flow architectures. Since our proposed masking scheme can be used as a drop-in replacement, we compare the different graphical models using this scheme to encode domain knowledge in GRF, GNF and SCCNF.

Table 6: Flow architectures. The hidden layer width and choice of activation function are the settings used for the conditioner neural network for GNF-A and GNF-M, the residual block for GRF, and the main flow transformation neural network for SCCNF, respectively.

| BN | Flow | Number of parameters | Number of flow steps* | Hidden layer width | Activation function |
|---|---|---|---|---|---|
| Arithmetic circuit | GNF-A$_S$ | 4416 | 4 | 200 | ReLU |
| | GNF-M$_S$ | 4552 | 2 | 50 | ReLU |
| | SCCNF$_S$ | 4621 | 3 | 140 | Tanh |
| | GRF$_S$ | 4296 | 8 | 125 | LipMish |
| | GNF-A$_L$ | 14796 | 9 | 300 | ReLU |
| | GNF-M$_L$ | 14508 | 4 | 125 | ReLU |
| | SCCNF$_L$ | 13312 | 5 | 150 | Tanh |
| | GRF$_L$ | 14518 | 17 | 200 | LipMish |
| Tree | GNF-A$_S$ | 4832 | 4 | 250 | ReLU |
| | GNF-M$_S$ | 4934 | 2 | 75 | ReLU |
| | SCCNF$_S$ | 4415 | 3 | 125 | Tanh |
| | GRF$_S$ | 4576 | 8 | 125 | LipMish |
| | GNF-A$_L$ | 14490 | 10 | 300 | ReLU |
| | GNF-M$_L$ | 14208 | 4 | 150 | ReLU |
| | SCCNF$_L$ | 13828 | 5 | 140 | Tanh |
| | GRF$_L$ | 14625 | 15 | 215 | LipMish |
| Protein | GNF-A$_S$ | 4812 | 4 | 175 | ReLU |
| | GNF-M$_S$ | 4897 | 1 | 150 | ReLU |
| | SCCNF$_S$ | 4788 | 3 | 150 | Tanh |
| | GRF$_S$ | 4788 | 9 | 100 | LipMish |
| | GNF-A$_L$ | 13788 | 9 | 225 | ReLU |
| | GNF-M$_L$ | 14691 | 3 | 150 | ReLU |
| | SCCNF$_L$ | 14765 | 5 | 170 | Tanh |
| | GRF$_L$ | 1896 | 28 | 100 | LipMish |
| EColi | GNF-A$_S$ | 4852 | 4 | 140 | ReLU |
| | GNF-M$_S$ | 4664 | 1 | 100 | ReLU |
| | SCCNF$_S$ | 4699 | 3 | 250 | Tanh |
| | GRF$_S$ | 4347 | 9 | 100 | LipMish |
| | GNF-A$_L$ | 14472 | 9 | 190 | ReLU |
| | GNF-M$_L$ | 13992 | 3 | 100 | ReLU |
| | SCCNF$_L$ | 13165 | 5 | 300 | Tanh |
| | GRF$_L$ | 14944 | 16 | 200 | LipMish |
| MEHRA | GNF-A$_S$ | 4760 | 4 | 150 | ReLU |
| | GNF-M$_S$ | 4976 | 1 | 125 | ReLU |
| | SCCNF$_S$ | 4355 | 3 | 150 | Tanh |
| | GRF$_S$ | 4797 | 9 | 125 | LipMish |
| | GNF-A$_L$ | 14220 | 9 | 200 | ReLU |
| | GNF-M$_L$ | 14928 | 3 | 125 | ReLU |
| | SCCNF$_L$ | 14214 | 5 | 175 | Tanh |
| | GRF$_L$ | 14382 | 17 | 200 | LipMish |

*For SCCNF, this instead refers to the number of layers in the neural network.

**SIReN-VAE**   Table 7 provides the network architectures used in the SIReN-VAE experiments. The models had the same network architectures for all of the datasets, except Arth. For Arth, the hidden layers of the neural networks had a width of 200 units, not 100. The weights of the decoder neural network and all residual blocks were masked according to the given BN structure.

When applying warm-up, a default warm-up period of 100 epochs was used in all cases. We used 32 importance-weighted samples for the Arithmetic Circuit 2 and MEHRA datasets when employing the importance-weighted objective, and only 8 for the Gaussian BNs, EColi and Arth, since further increasing the number of samples did not lead to notably better results.

Table 7: Model architectures used for the graphical datasets. All GRFs have 5 transformation steps, with each residual block having the same architecture as given below. D indicates the number of observed variables, and K the number of latent variables associated with each dataset's BN. Although not indicated here, the weight matrices of the linear layers in the decoder neural network and residual blocks of SIReN-VAE are masked according to the encoded BN structure. For $GRF_g$, the input dimension of the first linear layer of each residual block is larger to accommodate conditioning on the observation.

| Model | | Architecture |
|---|---|---|
| VAE | Encoder | Linear(D,100) $\rightarrow$ ReLU $\rightarrow$ Linear(100,K$\times$2) |
| | Decoder | Linear(K,100) $\rightarrow$ ReLU $\rightarrow$ Linear(100,D$\times$2) |
| SIReN-VAE | Encoder | $GRF_g$ |
| | Decoder | $GRF_n$ $\rightarrow$ Linear(K,100) $\rightarrow$ ReLU $\rightarrow$ Linear(100,D$\times$2) |
| | Residual Block | Linear(K(+D),100) $\rightarrow$ LipMish $\rightarrow$ Linear(100,K) |

## B.3   Additional Results

### B.3.1   Flow Inversion

Figure 14 gives the inversion performance of the Banach and Newton-like (Song et al., 2019) fixed-point approaches on the Arithmetic Circuit dataset. Table 8 summarizes the inversion performance of the different flow models on the rest of the datasets, with Figures 15 to 18 plotting the reconstruction error of GNF-M and GRF for different values of $\alpha$ in (7), when varying the number of iterations used at each step while inverting these flows.

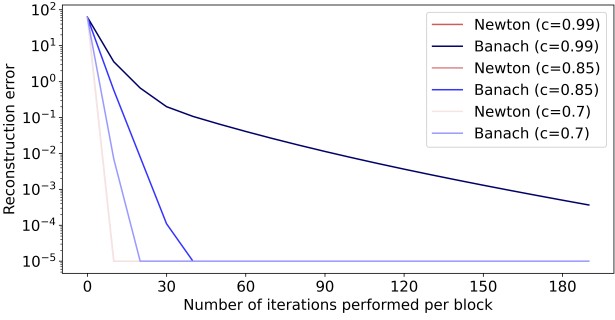

Figure 14: Using the Newton-like inversion procedure of Equation (7) requires far fewer iterations per block to accurately invert a GRF than using the Banach fixed-point approach. The plot shows the average reconstruction error (log-scale) for 100 samples from the Arithmetic Circuit test set. Note that all the plots for the Newton-like inversion procedure, corresponding to different values of the Lipschitz-bound hyperparameter $c$, overlap. Similar results were observed for the other datasets as well.

Table 8: Comparison of the inversion performance for the different flow models on 100 test data points from the various datasets. Bold indicates the best results in each column. $N$ and $\alpha$ are not applicable for SCCNF. Ranges indicate different optimal settings for $N$ and $\alpha$ for different data points. Inversion time is measured for the smallest $N \leq 50$ that allowed the most data points in the batch to have a reconstruction error of less than $10^{-4}$ and is the time taken to invert the entire batch.

| BN | Flow | Small Budget | | | | Large Budget | | | |
|---|---|---|---|---|---|---|---|---|---|
| | | Converged within 50 steps | $N$ | $\alpha$ | Inversion time (ms) | Converged within 50 steps | $N$ | $\alpha$ | Inversion time (ms) |
| Arithmetic Circuit | GNF-M | 99 | 4–42 | 0.3–1.1 | 226.17 | **100** | 5–12 | 1.0 | 141.83 |
| | SCCNF | 82 | — | — | 294.63 | 97 | — | — | 540.38 |
| | GRF | **100** | 4–5 | 1.0 | **50.15** | 100 | 3–4 | 1.0 | **91.88** |
| Protein | GNF-M | 97 | 9–50 | 0.5–1.4 | 145.88 | **100** | 5–32 | 0.8–1.2 | 268.28 |
| | SCCNF | 93 | — | — | 186.08 | 81 | — | — | 890.38 |
| | GRF | **100** | 5–7 | 0.9–1.0 | **71.90** | 100 | 4–8 | 0.9–1.0 | **265.23** |
| EColi | GNF-M | **100** | 7–45 | 0.4–1.0 | 182.40 | **100** | 6–8 | 1.0 | **98.90** |
| | SCCNF | 23 | — | — | 121.66 | 6 | — | — | 516.53 |
| | GRF | **100** | 5–6 | 1.0 | **57.91** | 100 | 4–5 | 1.0 | 101.90 |
| MEHRA | GNF-M | **100** | 3–11 | 0.9–1.0 | **32.91** | **100** | 3–7 | 0.8–1.1 | **62.81** |
| | SCCNF | **100** | — | — | 94.27 | 99 | — | — | 190.92 |
| | GRF | **100** | 3–4 | 1.0 | 48.15 | 100 | 3–4 | 1.0 | 89.69 |

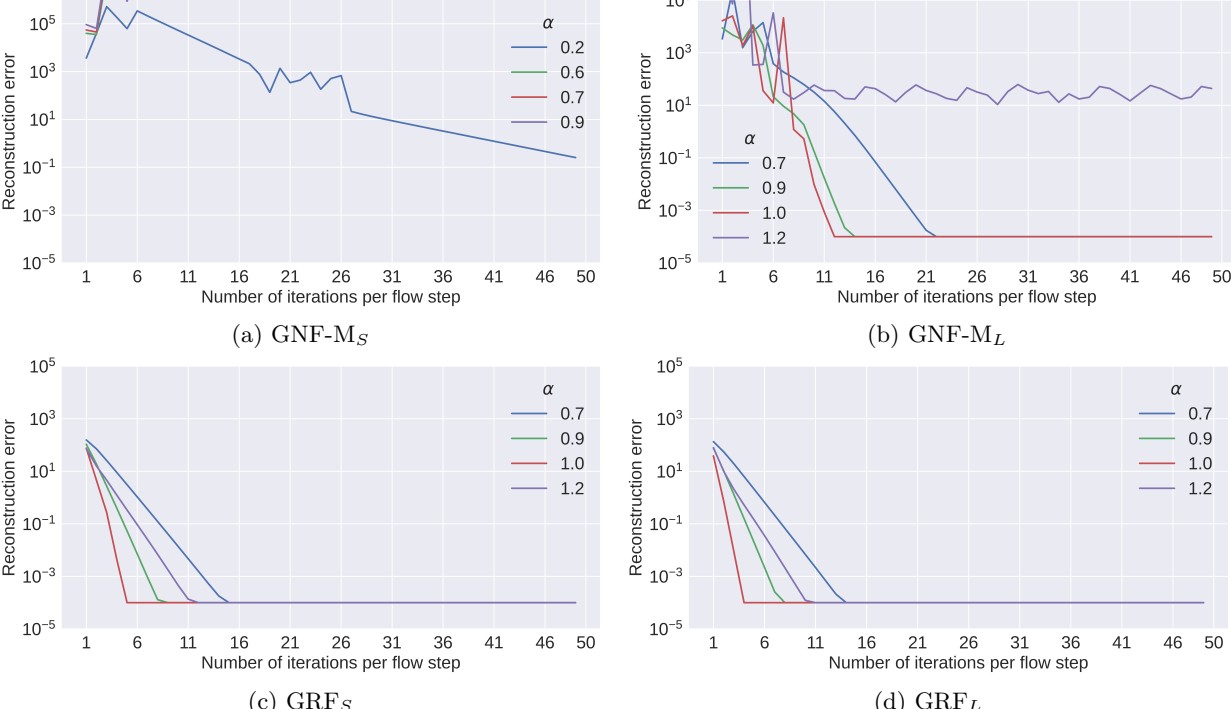

(a) GNF-M$_S$

(b) GNF-M$_L$

(c) GRF$_S$

(d) GRF$_L$

Figure 15: Reconstruction error achieved when inverting GNF-M and GRF on 100 test samples from the Arithmetic Circuit dataset. The reconstruction error is plotted as a function of the number of iterations used to invert each flow step, for different values of the step-size, $\alpha$.

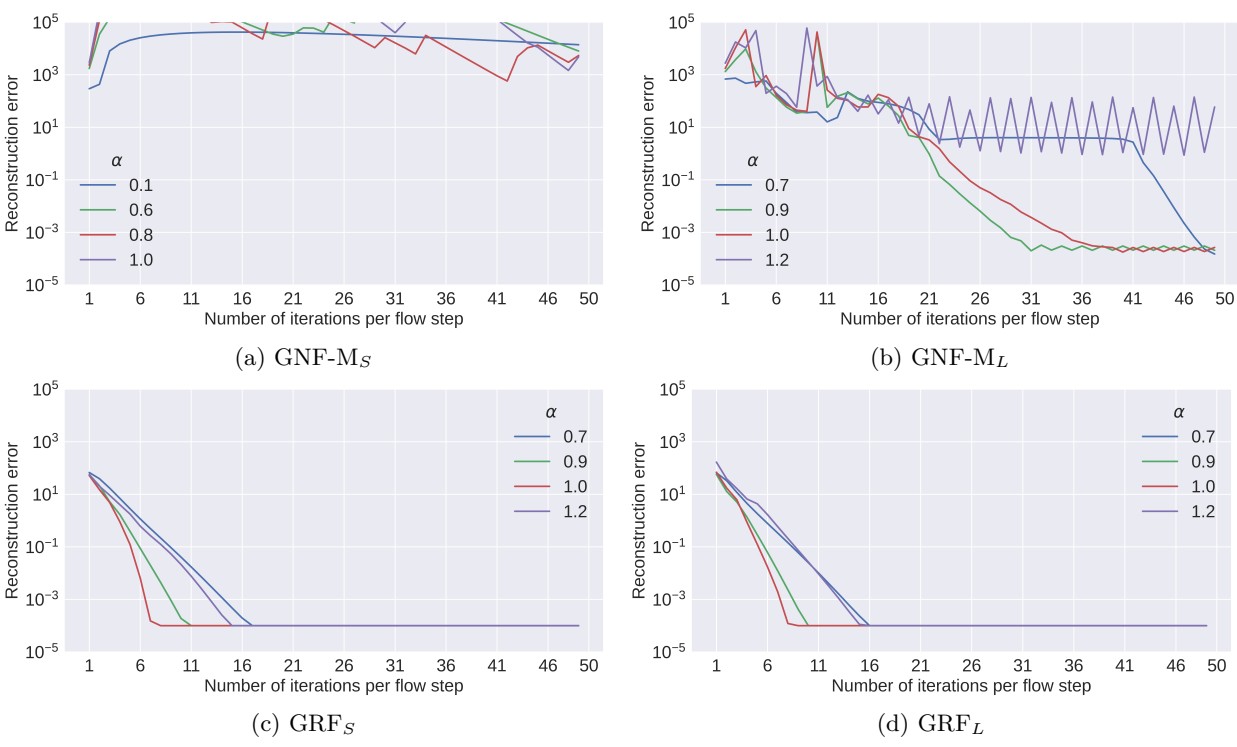

Figure 16: Reconstruction error when inverting GNF-M and GRF on samples from the Protein dataset.

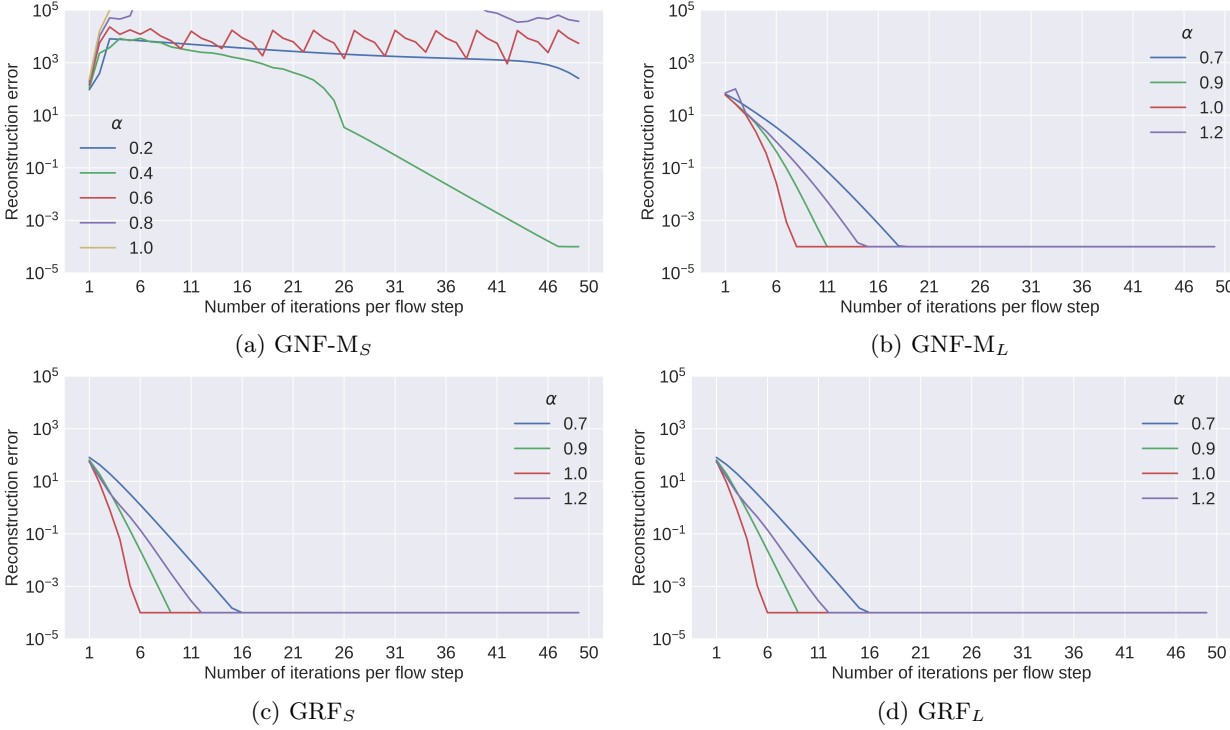

Figure 17: Reconstruction error when inverting GNF-M and GRF on samples from the EColi dataset.

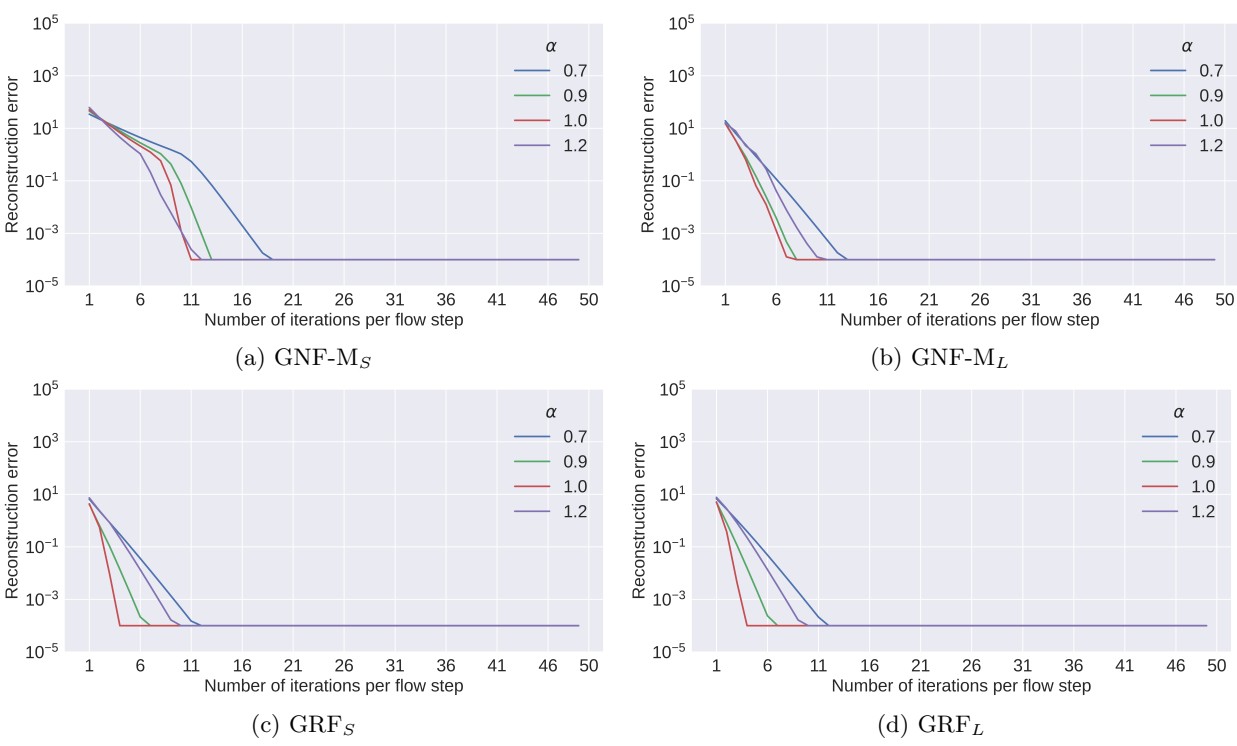

Figure 18: Reconstruction error when inverting GNF-M and GRF on samples from the MEHRA dataset.

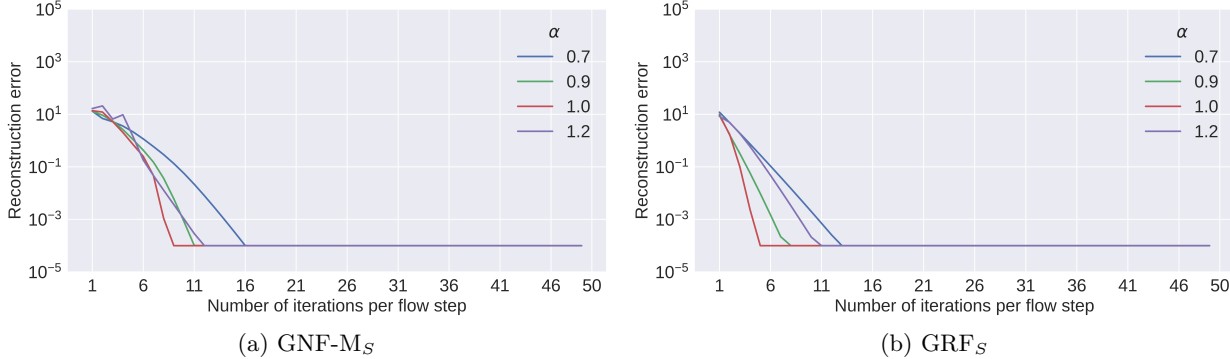

Figure 19: Reconstruction error when inverting GNF-M and GRF on samples from the Tree dataset.

### B.3.2 Performance of the LipMish Activation Function

Table 9 gives the performance results of GRF on the different datasets when using either the LipSwish or LipMish activation functions. See Figure 20 for a comparison of the first and second derivatives of these two activation functions. Apart from the activation function, the model architectures are the same as for $GRF_S$ and $GRF_L$ as detailed in Table 6. We also determine the negative log-likelihoods achieved by GRF as a function of the flow depth, when using different residual block activations, see Figure 21. We see that LipMish typically slightly outperforms LipSwish. Figure 21 further supports this, with LipMish performing on par and often better than other activations.

Table 9: Density estimation and variational inference performance of $GRF_S$ and $GRF_L$ when using either the LipMish or LipSwish activation function. Lower is better in all cases. Note that the real-world Protein and MEHRA datasets were not used for the inference tasks.

| BN | Flow | Density estimation (NLL) | | Inference (ELBO) | |
|---|---|---|---|---|---|
| | | LipSwish | LipMish | LipSwish | LipMish |
| Arithmetic | $GRF_S$ | $1.270_{\pm 0.03}$ | $\mathbf{1.248_{\pm 0.01}}$ | $4.219_{\pm 0.18}$ | $\mathbf{4.194_{\pm 0.19}}$ |
| Circuit | $GRF_L$ | $\mathbf{1.107_{\pm 0.01}}$ | $1.110_{\pm 0.01}$ | $3.766_{\pm 0.12}$ | $\mathbf{3.713_{\pm 0.14}}$ |
| Tree | $GRF_S$ | $8.649_{\pm \Delta}$ | $\mathbf{8.642_{\pm 0.01}}$ | $1.739_{\pm \Delta}$ | $\mathbf{1.738_{\pm \Delta}}$ |
| | $GRF_L$ | $8.649_{\pm \Delta}$ | $\mathbf{8.645_{\pm \Delta}}$ | $\mathbf{1.705_{\pm \Delta}}$ | $\mathbf{1.705_{\pm \Delta}}$ |
| Protein | $GRF_S$ | $-5.230_{\pm 0.02}$ | $\mathbf{-5.265_{\pm 0.01}}$ | — | — |
| | $GRF_L$ | $-6.035_{\pm 0.07}$ | $\mathbf{-6.111_{\pm 0.01}}$ | — | — |
| EColi | $GRF_S$ | $40.062_{\pm \Delta}$ | $\mathbf{40.059_{\pm \Delta}}$ | $34.986_{\pm 0.04}$ | $\mathbf{34.964_{\pm \Delta}}$ |
| | $GRF_L$ | $40.064_{\pm \Delta}$ | $\mathbf{40.062_{\pm \Delta}}$ | $\mathbf{34.962_{\pm \Delta}}$ | $34.963_{\pm \Delta}$ |
| MEHRA | $GRF_S$ | $11.665_{\pm 0.01}$ | $\mathbf{11.660_{\pm 0.02}}$ | — | — |
| | $GRF_L$ | $11.623_{\pm 0.04}$ | $\mathbf{11.612_{\pm 0.03}}$ | — | — |

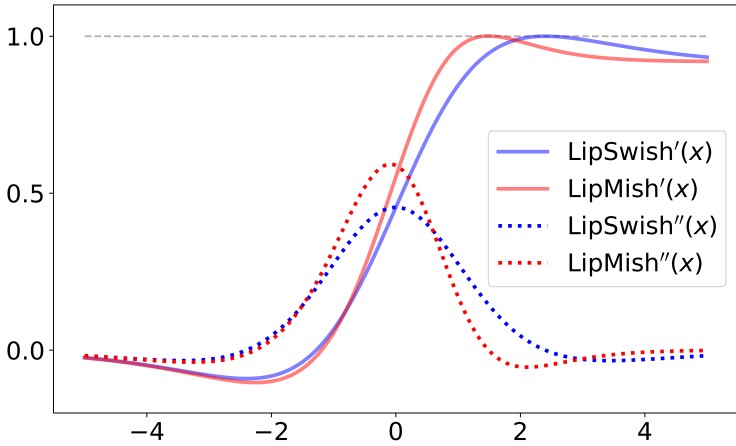

Figure 20: Comparison of the first and second derivatives of the LipSwish (Chen et al., 2019) and the proposed LipMish activation functions.

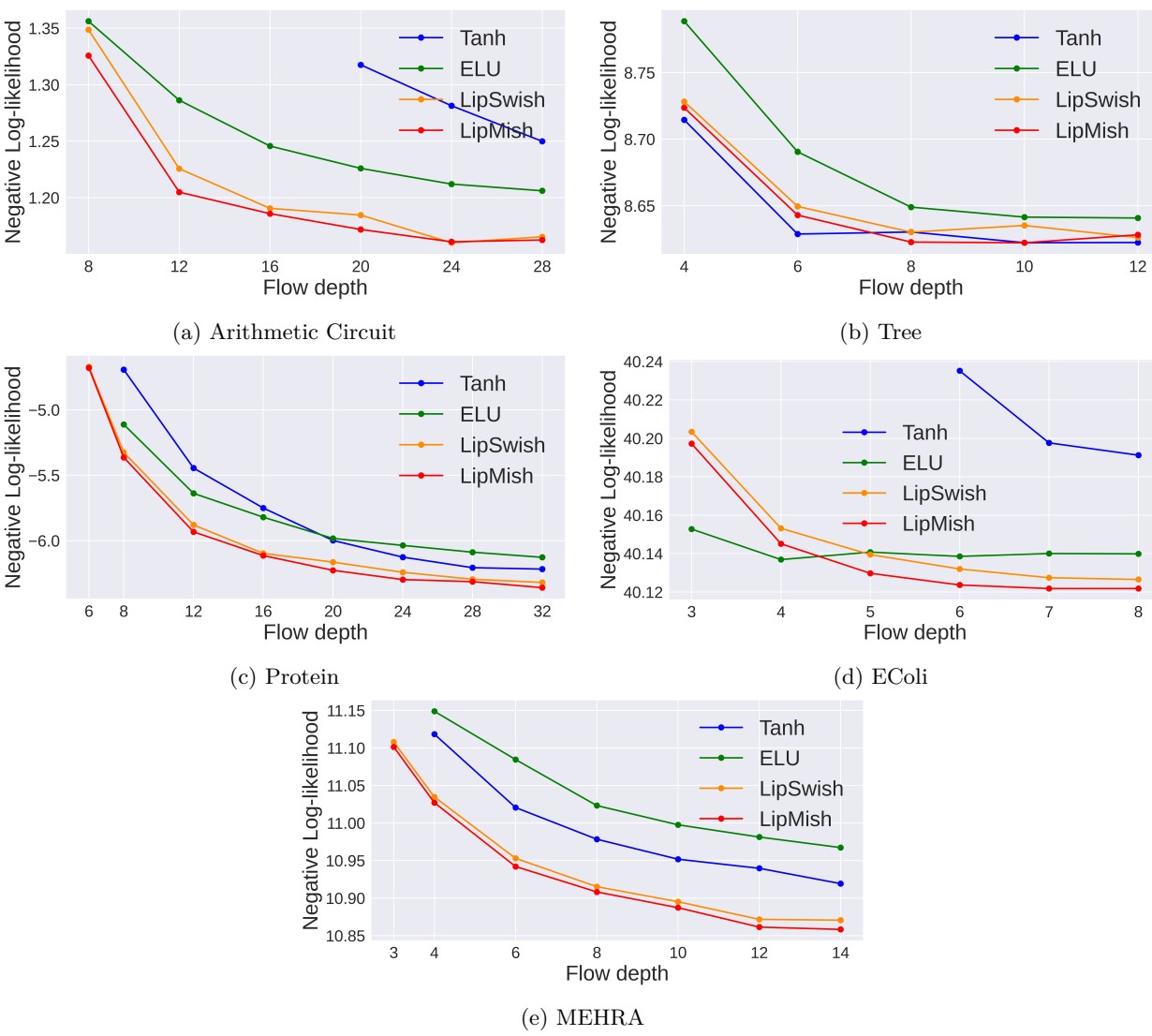

Figure 21: Negative log-likelihood achieved by GRF models for varying flow depths and different residual block activation functions (tanh, ELU, LipSwish and LipMish) on each of the datasets. Lower is better.

### B.3.3 Dependency Structure Induced by GRFs

We provide empirical results supporting our argument that even though information leakage may occur in a GRF between variables and their ancestors, the flow architecture still provides a strong enough inductive bias to encourage the resulting distribution to respect the provided dependence structure. If the induced dependence structure fully corresponds to the transitive closure of the BN graph $\mathcal{G}$, denoted by $\mathrm{TC}(\mathcal{G})$, then another BN graph, $\mathcal{G}'$, where $\mathrm{TC}(\mathcal{G}') = \mathrm{TC}(\mathcal{G})$, should provide similar performance. We investigate this for the Tree, Protein and EColi datasets by constructing such alternative graphs. These $\mathcal{G}'$ graphs can be constructed by removing each edge in the original graph that does not change its transitive closure, and adding the same number of new edges that are in the transitive closure of $\mathcal{G}$, but not in $\mathcal{G}$ itself. The type of edge that can be removed without changing the transitive closure, is any edge between a child and one of its parents where this parent is also an ancestor of another one of the child's parents. The above construction ensures that $\mathcal{G}'$ has the same number of edges as $\mathcal{G}$, as well as a matching transitive closure, while specifying a different set of independence assumptions. We also compared using $\mathcal{G}$ and $\mathcal{G}'$ to using a graph $\mathcal{G}'_{\min}$, where *all* possible edges have been removed such that $\mathrm{TC}(\mathcal{G}'_{\min}) = \mathrm{TC}(\mathcal{G})$. Since the Arithmetic

Circuit and MEHRA BNs do not have any such edges that can be used to modify the dependence structure, we did not consider them here. Figures 22 and 23 show the graphs $\mathcal{G}$, $\mathcal{G}'$ and $\mathcal{G}'_{\min}$ used for the Tree and Protein datasets, respectively. Table 10 provides the modelling performance of the small and large GRF models when encoding either $\mathcal{G}$, $\mathcal{G}'$ and $\mathcal{G}'_{\min}$ for density estimation. The architectures of these models are the same as given in Table 6.

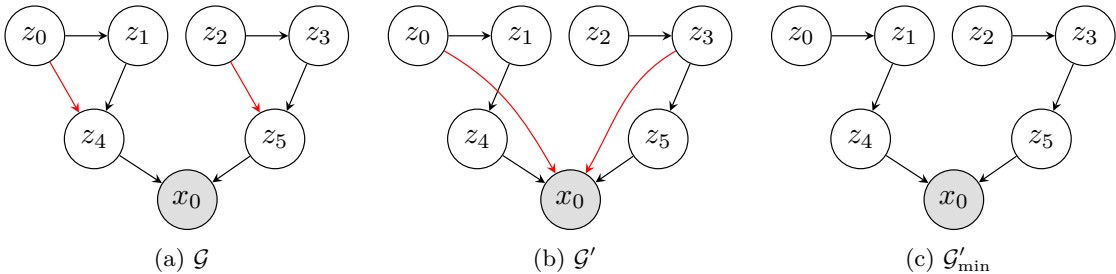

Figure 22: Illustration of the BN graphs encoded into the GRFs presented in Table 10 for the Tree dataset. Red edges in (a) the true BN graph were removed and placed between different child-ancestors pairs to create (b) a different graph with the same transitive closure as (a), or removed entirely to obtain (c) a graph with the minimum number of edges that still has the same transitive closure as (a).

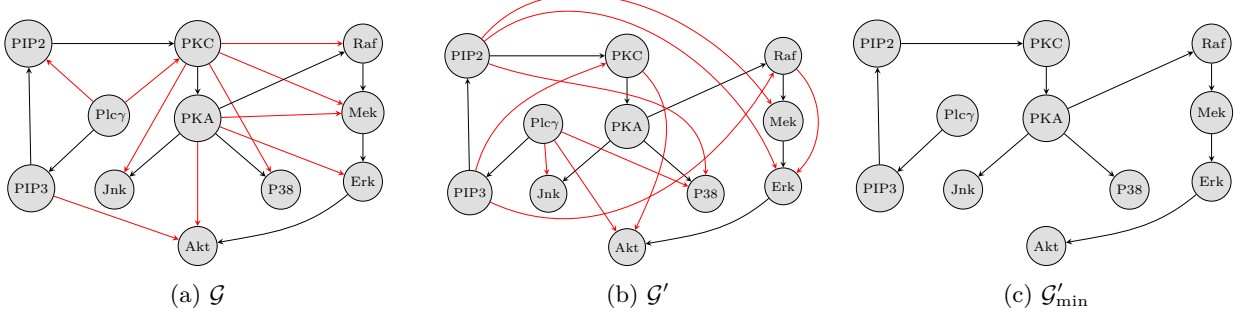

Figure 23: Illustration of the BN graphs encoded into the GRFs presented in Table 10 for the Protein dataset.

Table 10: Density estimation performance of $\text{GRF}_S$ and $\text{GRF}_L$ when encoding either the true BN graph, $\mathcal{G}$, or alternative graphs $\mathcal{G}'$ and $\mathcal{G}'_{\min}$ where $\text{TC}(\mathcal{G}) = \text{TC}(\mathcal{G}') = \text{TC}(\mathcal{G}'_{\min})$ still holds. TC is the transitive closure of a graph. For each BN, we provide the number of edges that were either removed or placed between different child-ancestor pairs to obtain the graphs $\mathcal{G}'_{\min}$ and $\mathcal{G}'$, respectively. Each entry corresponds to the average over the test set for five independent runs, with standard deviation given in the subscript. Lower is better in all cases. The best performance in each group is indicated with bold.

| BN | #Edges | Flow | Density estimation (NLL) | | |
| --- | --- | --- | --- | --- | --- |
| | | | $\mathcal{G}$ | $\mathcal{G}'$ | $\mathcal{G}'_{\min}$ |
| Protein | 9 | $\text{GRF}_S$ | $\mathbf{-5.26_{\pm 0.01}}$ | $-4.87_{\pm 0.04}$ | $-4.41_{\pm 0.11}$ |
| | | $\text{GRF}_L$ | $\mathbf{-6.12_{\pm 0.01}}$ | $-5.85_{\pm 0.13}$ | $-5.65_{\pm 0.09}$ |
| EColi | 12 | $\text{GRF}_S$ | $\mathbf{40.06_{\pm 0.00}}$ | $40.10_{\pm 0.01}$ | $40.27_{\pm 0.10}$ |
| | | $\text{GRF}_L$ | $\mathbf{40.06_{\pm 0.00}}$ | $40.07_{\pm 0.00}$ | $40.10_{\pm 0.02}$ |
| Tree | 2 | $\text{GRF}_S$ | $\mathbf{8.64_{\pm 0.01}}$ | $8.66_{\pm 0.00}$ | $8.65_{\pm 0.00}$ |
| | | $\text{GRF}_L$ | $\mathbf{8.64_{\pm 0.00}}$ | $8.66_{\pm 0.00}$ | $8.65_{\pm 0.00}$ |

The results show that models encoding the original dependency structure provided the best modelling performance in all cases. This is most noticeable for the real-world Protein dataset. Even for the simple Gaussian EColi BN and for the Tree BN where only two edges had been changed, the original BN graph still provided the best performance. We suspect therefore that even though information leakage may occur, the direct dependencies encoded by the flow given the BN, still play a more important role in informing the resulting distribution. While the GRF formulation does allow dependencies corresponding to the transitive closure of the encoded BN graph, we expect the forms of the dependencies on variables beyond the underlying graph to be quite constrained, making learning more sophisticated dependencies between vertices not connected in the underlying BN more difficult. Although future work should investigate this issue in more detail, the above results help to motivate our use of GRFs.

### B.3.4   Addressing Posterior Collapse

Here, we provide the performance results of SIReN-VAE$_{\text{True}}$ on the full training set when employing different combinations of posterior collapse mitigation techniques. Table 11 expands the results presented in Section 5.2.2. In addition to warm-up and importance weighted objectives, we also considered using a lower-variance gradient estimator for optimization which is expected to reduce the likelihood that posterior collapse occurs (Melis et al., 2022). Specifically, we employed the doubly-reparameterized gradient (DReG) estimator of Tucker et al. (2018). As seen in Table 11, using this lower-variance gradient estimator typically resulted in the model achieving a better negative log-evidence as well as having fewer collapsed latent variables. However, using DReG requires inverting the GRF in the variational posterior (Vaitl et al., 2022), which both slows down training and results in higher memory usage (since the computational graphs of the fixed-point inversion iterations need to be stored to facilitate backpropagation). It is therefore not clear how worthwhile the use of DReG to improve results is, given this additional cost.

Table 11: Effect of applying different combinations of posterior collapse mitigation techniques, namely: warm-up (WU), importance weighted objectives (IWAE) and doubly-reparameterized gradient estimates (DReG). Warm-up is applied to either all latents ($WU_{all}$), or to a selected subset that is prone to collapse ($WU_{select}$). The performance metrics are the same as for Table 4.

| BN | Model | $-\log p(\mathbf{x})$ | #Inactive Units |
|---|---|---|---|
| Arithmetic Circuit 2 | $SIReN\text{-}VAE_{True}$ | $10.03_{\pm 0.01}$ | $2.00_{\pm 0.00}$ |
| | $SIReN\text{-}IWAE_{True}$ | $9.86_{\pm 0.04}$ | $1.00_{\pm 0.00}$ |
| | $SIReN\text{-}IWAE_{True}+DReG$ | $\mathbf{9.80_{\pm 0.02}}$ | $1.00_{\pm 0.00}$ |
| | $SIReN\text{-}VAE_{True}+WU_{all}$ | $10.11_{\pm 0.08}$ | $1.00_{\pm 0.00}$ |
| | $SIReN\text{-}IWAE_{True}+WU_{all}$ | $9.86_{\pm 0.08}$ | $1.00_{\pm 0.00}$ |
| | $SIReN\text{-}IWAE_{True}+DReG+WU_{all}$ | $9.88_{\pm 0.06}$ | $1.00_{\pm 0.00}$ |
| | $SIReN\text{-}VAE_{True}+WU_{select}$ | $9.82_{\pm 0.02}$ | $\mathbf{0.00_{\pm 0.00}}$ |
| | $SIReN\text{-}IWAE_{True}+WU_{select}$ | $\mathbf{9.80_{\pm \Delta}}$ | $\mathbf{0.00_{\pm 0.00}}$ |
| | $SIReN\text{-}IWAE_{True}+DReG+WU_{select}$ | $\mathbf{9.80_{\pm 0.02}}$ | $\mathbf{0.00_{\pm 0.00}}$ |
| EColi | $SIReN\text{-}VAE_{True}$ | $34.99_{\pm 0.01}$ | $\mathbf{0.00_{\pm 0.00}}$ |
| | $SIReN\text{-}IWAE_{True}$ | $\mathbf{34.98_{\pm \Delta}}$ | $\mathbf{0.00_{\pm 0.00}}$ |
| | $SIReN\text{-}IWAE_{True}+DReG$ | $\mathbf{34.98_{\pm 0.01}}$ | $\mathbf{0.00_{\pm 0.00}}$ |
| | $SIReN\text{-}VAE_{True}+WU_{all}$ | $34.99_{\pm 0.01}$ | $\mathbf{0.00_{\pm 0.00}}$ |
| | $SIReN\text{-}IWAE_{True}+WU_{all}$ | $34.99_{\pm 0.01}$ | $\mathbf{0.00_{\pm 0.00}}$ |
| | $SIReN\text{-}IWAE_{True}+DReG+WU_{all}$ | $34.99_{\pm \Delta}$ | $\mathbf{0.00_{\pm 0.00}}$ |
| Arth | $SIReN\text{-}VAE_{True}$ | $37.73_{\pm 0.05}$ | $14.80_{\pm 0.33}$ |
| | $SIReN\text{-}IWAE_{True}$ | $37.49_{\pm 0.26}$ | $11.00_{\pm 1.10}$ |
| | $SIReN\text{-}IWAE_{True}+DReG$ | $37.30_{\pm 0.53}$ | $6.00_{\pm 1.41}$ |
| | $SIReN\text{-}VAE_{True}+WU_{all}$ | $37.65_{\pm 0.08}$ | $11.20_{\pm 0.98}$ |
| | $SIReN\text{-}IWAE_{True}+WU_{all}$ | $37.48_{\pm 0.47}$ | $10.20_{\pm 1.33}$ |
| | $SIReN\text{-}IWAE_{True}+DReG+WU_{all}$ | $\mathbf{37.27_{\pm 0.54}}$ | $7.80_{\pm 1.33}$ |
| | $SIReN\text{-}VAE_{True}+WU_{select}$ | $37.71_{\pm 0.03}$ | $9.80_{\pm 2.64}$ |
| | $SIReN\text{-}IWAE_{True}+WU_{select}$ | $37.42_{\pm 0.24}$ | $7.00_{\pm 2.28}$ |
| | $SIReN\text{-}IWAE_{True}+DReG+WU_{select}$ | $37.43_{\pm 0.22}$ | $\mathbf{1.80_{\pm 0.98}}$ |
| MEHRA | $SIReN\text{-}VAE_{True}$ | $8.37_{\pm 0.06}$ | $\mathbf{0.00_{\pm 0.00}}$ |
| | $SIReN\text{-}IWAE_{True}$ | $8.19_{\pm 0.06}$ | $\mathbf{0.00_{\pm 0.00}}$ |
| | $SIReN\text{-}IWAE_{True}+DReG$ | $\mathbf{8.08_{\pm 0.02}}$ | $\mathbf{0.00_{\pm 0.00}}$ |
| | $SIReN\text{-}VAE_{True}+WU_{all}$ | $8.66_{\pm 0.26}$ | $\mathbf{0.00_{\pm 0.00}}$ |
| | $SIReN\text{-}IWAE_{True}+WU_{all}$ | $8.39_{\pm 0.24}$ | $\mathbf{0.00_{\pm 0.00}}$ |
| | $SIReN\text{-}IWAE_{True}+DReG+WU_{all}$ | $8.23_{\pm 0.08}$ | $\mathbf{0.00_{\pm 0.00}}$ |

