# OpenReview forum: "Integrating Bayesian Network Structure into Residual Flows and Variational Autoencoders"
_TMLR — Accepted by TMLR_

### Review · Reviewer_RTpd · 2023-02-01

**Summary Of Contributions:**

The paper focuses on incorporating a structure into random variables in a form of a Bayesian network (BN) to flow-based models and Variational Auto-Encoders (VAEs). For flow-based models, the idea is to mask weight matrices in such a way that the BN is retained. For VAEs, the latent space if factorized according to a given BN. In a series of experiments, the proposed idea seems to be preferable over other methods, especially for higher dimensional problems.

**Audience:**

Yes

**Broader Impact Concerns:**

No concerns.

**Claims And Evidence:**

Yes

**Requested Changes:**

R1: A better presentation and discussion of the proposed method (i.e., how to mask). See WQ1.
R2: Additional baseline (e.g., residual flows). See WQ2.

**Strengths And Weaknesses:**

*Strengths*:
+ The idea of incorporating a structure into (generally speaking) latent-variable models is interesting and promising.
+ The paper proposes a new flow-based model, dubbed Graphical Residual Flows (GRFs), which takes into account a BN-based structure.
+ The paper proposes a new VAE model, dubbed SiReN (structured invertible residual network) VAEs, which use a BN to form the latent space.
+ The experiments introduce some evidence for using the proposed methods.

*Weaknesses*:
- WQ1: The paper heavily builds upon two papers: (Wehenkel & Louppe, 2021) and (Weilbach et al., 2020). In such a situation, it is very important to highlight all differences and new ideas. I miss an example of how to formulate masks to introduce a BN. At the moment, it is quite hard to see the difference between previous papers and the proposed method. Figure 1 tries to do that but I miss a more in-depth presentation and discussion.
- WQ2: The experiments are interesting, however, I miss a good baseline. It would be greatly beneficial to run vanilla residual flows and report their performance. It would be more convincing if the flows with a BN outperforms this baseline.
- WQ3: In Table 3, why does SIReN-VAE perform sometimes worse (especially for the full training data) than SIReN-VAE-FC or VAE?

*Open questions*:
- OQ1: Would it be possible to extend the proposed idea to invertible DenseNet flows? (Perugachi-Diaz, Y., Tomczak, J., & Bhulai, S. (2021). Invertible densenets with concatenated lipswish. Advances in Neural Information Processing Systems, 34, 17246-17257.)
- OQ2: Would it make sense to do a two-stage training of SIReN-VAEs, more along (Dai, B., & Wipf, D. (2019). Diagnosing and enhancing VAE models. arXiv preprint arXiv:1903.05789). I can imagine that learning the variational posterior together with the GRF-based prior may not be beneficial from the optimization perspective. First training with a "naive" prior like standard Gaussian, and then turning on a more elaborated prior may result in even better results.

---

### Review · Reviewer_1tzL · 2023-02-15

**Summary Of Contributions:**

This paper introduces a new normalizing flow architecture that combines Bayesian Network structures, such as in graphical normalizing flows (GNFs) and SCCNF with invertible residual networks. The new architecture is naturally more stable for inversion and perform similarly to existing NFs embedding graphical constraints.
The authors show the architecture is effective in scarce data settings if we can hypothesise a good Bayesian network structure.

**Audience:**

Yes

**Broader Impact Concerns:**

I do not see any broader impact concerns.

**Claims And Evidence:**

No

**Requested Changes:**

- I would recommend to push further the experiments with various degree of mistakes in the graphical model used. A thorough analysis would provide valuable insights as when we can hope to gain by embedding graphical constraints which, in many cases, will not correspond to the "true" independencies.
- Cover better the topic of independencies in BNs.
- Figure 3 should be cleaned.
- Provide an experiments where a VAE has really a good reason of being there, e.g., if the observed signal is an image. I would be happy to iterate a bit with the authors to find something that is really nice here. In my opinion, this is an important aspects that could really improve the impact of this publication.
- Improve the discussion on failure points of the approach. For instance, learning with such constraints on the factorisation can lead to very complicated 1d distributions that hard to model with neural nets whereas a simpler architecture would work well.
- For the experiment with the VAE, also provide the numbers obtained with a simpler NFs (e.g. a 3-layer MAF or UMNN-MAF)
- Explain how you compute the MI for Figure 5.
- Plot the non-standard LipMIsh activation function.
- A discussion of strategies to apply when only partial knowledge of a good graph is known would be interesting.

**Strengths And Weaknesses:**

Overall I find this work interesting and would think TMLR is good venue it. However, I have some concerns I would like to see addressed before voting for acceptance. Providing a public implementation of this work is also very important in my opinion.

For now, I would attribute a "no" to the claims and evidence but I am quite convinced this should become a yes within the scope of a discussion with the authors and a few additional experiments.

# Weaknesses:
- I find the title misleading as there already exist NFs architectures integrating Bayesian networks.
- The contribution mainly builds on existing work (I-resnets and GNFs) and may seem incremental.
- The paper is lacking a good covering of independence in Bayesian networks whereas this is the key point in assuming a Bayesian Network structure.
- Some experiments could be pushed further in order to provide better insights regarding the value of the proposed methods (see requested changes).
- It is difficult to decipher what exactly is the key contribution here. To add structure within VAEs? To combine iresnet with graphical constraints? I think this should be clarified.
- As the technical contribution is somehow incremental, I would expect a better evaluation of the interest of incorporating structure into VAEs. To me, the current paper is not making this point unarguable although I am convinced this is indeed a good idea in many cases.

# Strengths:
- Building a better alternative of SCCNF and GNF is an interesting research direction as many deep probabilistic models are missing a point by not exploiting domain knowledge about independence between the random variables modelled. In particular, embedding the Graphical structure directly in the neuron's connectivity, similarly to MAF and MADE is a nice to have.
- The paper reads well.
- The methods works well compared to GNF and SCCNF, this is a really important point.
- A lot of effort has been made to create a method that is stable numerically. The experiments and discussion about the posterior collapse are interesting.
- I also really liked the idea of using the structure inversion algorithm!

---

### Review · Reviewer_YbNy · 2023-02-18

**Summary Of Contributions:**

The paper tackle the issue of how to incorporate dependency structure (represented by a bayesian network) into the training of generative models (in particular, VAEs). The paper proposes to Graphical Residual Flow (GRF), which is a graphical normalizing flow whose inverse can be accurately computed, to encode the predefined dependency structure over the latent and observed variables. To preserve the dependency information, a masking mechanism is used to ensure that the output at each dimension is only a function of the itself and its parents. Experimental results show that VAE equipped with GRF obtains better likelihood.

**Audience:**

Yes

**Claims And Evidence:**

Yes

**Requested Changes:**

As said in my comments, I would request authors to include introduction of some missing parts of the model in the main text, as well as a section that tells the motivation/application of this model.

**Strengths And Weaknesses:**

Strength:

1. The paper is very well written, with a high quality review of related literature and clear introduction of background knowledges. Graphical illustration is also very helpful.
2. The idea of introducing GRF to both the prior and variational posterior in order to model the dependency is interesting, and the success of it highly depends on the novel design of GRF which allows fast and accurate inverse computation.
3. The experimental results are comprehensive, with multiple synthetic and real datasets included. Results do show some improvement over baselines. Although the improvements are not very significant, as the author claimed, the method has its own advantage when the training set is small.

Weakness:

1. I think the masking mechanism is an important component of the model, and it should be clearly introduced in the main text rather than in appendix. As an exchange, some background information may be moved to the appendix.
2. While incorporating explicit information regarding dependency in training latent variable models is definitely an interesting problem, it would be better to introduce  more background/motivation for studying this problem. In particular, since in reality the explicit dependency is difficult to obtain (that's why so many people are studying causal inference), what are the potential application of this method? Is there any cases in real life that the model can be useful? These would let the reader better appreciate the work.

---

### Author Response · Authors · 2023-04-22
**Video link**

A video accompanying the paper can be found at: [https://youtu.be/QCNpBmD91IE](https://youtu.be/QCNpBmD91IE) .

---

### Decision · Action_Editors · 2023-03-21

**Recommendation:** Accept as is

**Comment:**

The paper already went through three revisions during the discussion with the reviewers. In these revisions, the authors expanded and clarified the content of the paper, added additional experiments, and addressed all reviewers' concerns. I don't think the paper needs an additional revision, so I'm happy to recommend acceptance as is.

To the authors: please don't forget to add the link to the code in the camera-ready version of the paper.


**Audience:**

The paper studies incorporating prior structure (expressible as a Bayesian network) into residual flows and, in extension, VAEs. This is of interest to TMLR's audience.

**Claims And Evidence:**

All three reviewers are satisfied that the paper's claims are supported by sufficient evidence.

Reviewer 1tzL expressed some concerns in their review initially, but these were addressed in the discussion that followed.